# In vivo crosslinking and effective 2D enrichment for proteome wide interactome studies
Philipp Bräuer[1], Laszlo Tirian[2], Fränze Müller [1], Karl Mechtler [1,2,3] ✉ & Manuel Matzinger [1] ✉

Cross-linking mass spectrometry has evolved as a powerful technique to study protein-protein interactions and to provide structural information. Low reaction efficiencies, and complex matrices lead to challenging system wide crosslink analysis. We improved and streamlined an Azide-A-DSBSO based in vivo crosslinking workflow employing two orthogonal effective enrichment steps: Affinity enrichment and size exclusion chromatography (SEC). Combined, they allow an effective enrichment of DSBSO containing peptides and remove the background of linear as well as mono-linked peptides. We found that the analysis of a single SEC fraction is effective to yield ~90% of all crosslinks, which is important whenever measurement time is limited, and sample throughput is crucial. Our workflow resulted in more than 5000 crosslinks from K562 cells and generated a comprehensive PPI network. From 393 PPI found within the nucleus, 56 are novel. We further show, that by applying DSBSO to nuclear extracts we yield more crosslinks on lower abundant proteins and showcase this on the DEAD-box RNA helicase DDX39B which is predominantly expressed in the nucleus. Our data indicates that DDX39B might be present in monomeric and dimeric forms together with DDX39A within the nuclear extracts analyzed.

Protein–protein interactions (PPIs) are fundamental to functional networks and biological processes. Studying the structural and dynamic organization of protein interaction networks in their native cellular environment is challenging. Traditional methods like co-immunoprecipitation, yeast two-hybrid, and proximity-labeling mass spectrometry (MS) might be limited by antibody availability and require cell lysis, which leads to a loss of the native environment and limits the detection of transient or weak interactions. High-resolution three-dimensional protein structure information is typically obtained by techniques such as X-ray crystallography, nuclear magnetic resonance spectroscopy, or cryo-electron microscopy, which require large amounts of purified proteins, again removing protein complexes from their native environment.

Chemical cross-linking mass spectrometry (XL-MS) has emerged as a complementary tool, providing low-resolution native structure information[1]. Recent advancements in crosslinker (XL) molecules, enrichment strategies, MS methods, and data interpretation software have highlighted the growing importance of XL-MS[2].

XL-MS can identify PPIs on a proteome-wide level in their native environment and is therefore of great importance for studying biological processes such as cellular signaling pathways and transient protein interactions. However, proteome-wide studies pose a challenge for data analysis (n² problem[3]) and have to deal with high sample complexity, dynamic abundance ranges, and low cross-linking efficiencies. The n² problem refers to the exponential increase in computational resources required as the number of peptide-peptide combinations for a crosslink search increases exponentially with the size of the database used for analysis.

To overcome the above challenges, the development of MS-cleavable XLs, which contain labile bonds that cleave upon collisional activation to produce characteristic signature ions, has significantly advanced the field[4]. In situ applications face additional difficulties, such as cell membrane penetration and high hydrolysis rates of XL molecules, further reducing reaction efficacy.

To address these challenges, we use disuccinimidyl bis-sulfoxide (DSBSO), an MS cleavable, enrichable linker that was shown to be surprisingly well membrane permeable[5] given the charged nitrogen atoms inside the net neutral azide residue. It was already successfully used for in vivo studies before[6,7]. We applied a streamlined sample preparation protocol, improving previously reported protocols[8,9] using complementary

[1]Research Institute of Molecular Pathology (IMP), Vienna BioCenter (VBC), Vienna, Austria. [2]Institute of Molecular Biotechnology (IMBA), Austrian Academy of Sciences, Vienna BioCenter (VBC), Vienna, Austria. [3]Gregor Mendel Institute of Molecular Plant Biology (GMI), Austrian Academy of Sciences, Vienna BioCenter (VBC), Vienna, Austria. ✉e-mail: karl.mechtler@imp.ac.at; manuel.matzinger@imp.ac.at

enrichment strategies to investigate the nuclear interactome in K562 cells. We further investigate interaction partners and align our experimentally found crosslinks to the predicted 3D structure of DDX39B. This DEAD-box RNA helicase, also known as UAP56, is primarily localized in the nucleus and plays a crucial role in pre-mRNA splicing, mRNA export, and genome stability maintenance. It is involved in a multitude of essential cellular processes, and its dysregulation is linked to various cancers, making it a potential target for therapeutic interventions[10,11].

## Results

### Optimizing enrichment workflow

To capitalize on the advantages of an azide-tagged linker such as DSBSO, we enriched cross-linked peptides on alkyne-functionalized beads. Since monolinked peptides, where one end of the crosslinker molecule is connected to a peptide while the other end is hydrolyzed in an aqueous environment, are co-enriched, a second orthogonal enrichment based on size exclusion was applied. We thereby adapted (a) our own previous work[8], and (b) a recent protocol from the Lan Huang lab[9] to yield a streamlined and improved workflow (Fig. 1).

For this, we first applied DSBSO (2 mM) to K562 cells, followed by lysis and filter-aided sample preparation (FASP). This was followed by the affinity enrichment step (**a**). Thus, we employed a copper-free strain-promoted alkyne azide cycloaddition click reaction using dibenzocyclooctyne (DBCO) to form a stable triazole covalently connecting crosslinked peptides to the bead material. Instead of using Sepharose-based bead material, as previously published[8], we coupled the DBCO group to magnetic beads, which allows for facilitated, more effective washing steps yielding improved crosslink numbers (Supplementary Fig. 1). We additionally decided to benchmark magnetic bead material from two different vendors: Cytiva and Cube Biotech. Both are functionalized with active N-hydroxy succinimide (NHS) ester groups, which we coupled to DBCO-Amine. Their density of active group ranges from 8 to 14 μmol/mL for the Cytiva to >15 μmol/mL for the Cube Biotech beads as specified by the manufacturers. To benchmark them

for the purpose of affinity enrichment of DSBSO-linked peptides, we chose purified, DSBSO-linked, and digested Cas9 as a model system. We spiked the crosslinked Cas9 peptides into a background of non-crosslinked HeLa peptides in a 1:80 (w:w) ratio, to mimic a more complex sample matrix. Of note, the relative abundance of non-crosslinked material in a real *in cellulo* experiment is even higher; hence, this is only the first step using a simple model system to evaluate enrichment performance. Our results showed that the average number of unique crosslink sites (XLs, referring to unique residue pairs within this study, inter-XL are inter-protein crosslink connections within this study) was improved by 169% using Cube Biotech beads and even by 314% using Cytiva and the purified protein as a sample. This is already showing that coupling to DBCO and enrichment per se worked as expected (Fig. 2A). Beads from Cytiva seemed to clearly outperform those from Cube Biotech in terms of reachable crosslink numbers, with no significant difference in the yielded background level of linear peptides (Fig. 2C).

After spiking, the yielded benefit from enrichment is even larger than on the purified protein alone, as no crosslinks and only very few monolinks could be identified within the added complex matrix of a whole cell digest (Fig. 2A, B). After a one-step affinity enrichment, though, >200 crosslinks could still be recovered. This shows general functionality of the applied enrichment but also potential limitations in terms of losses upon dilution of the reactive groups (of the crosslinked sample), as only 33–48% of the crosslink numbers reached after enrichment from the clean protein could be reached. In contrast to the results on the purified protein alone, within the more complex environment, both bead vendors performed at a very comparable level. We observed no significant difference by means of identified unique crosslinks, identified crosslink spectrum matches (CSMs, with CSMs referring to redundant matches found within this study), or co-enriched monolinked PSMs (referred to as redundant sequence matches found within this study) (Fig. 2A, B). The background of non-crosslinked, linear peptides that bound unspecifically seemed to be slightly, but not significantly, lowered for beads from Cube Biotech (Fig. 2C). Of note, the Cube

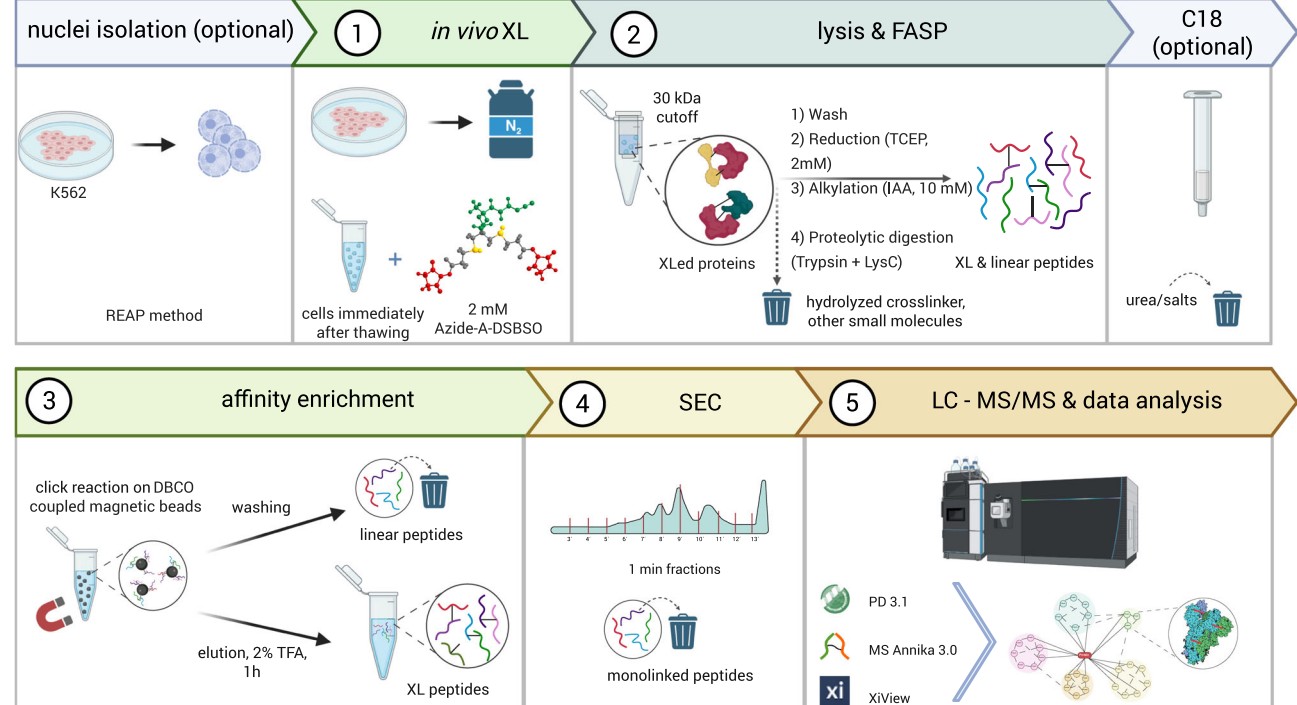

**Fig. 1 | Graphical workflow representation.** Cells or isolated nuclei are cross-linked using Azide-A-DSBSO (1) followed by lysis, reduction, alkylation, and digestion on a FASP filter (2). After an optional C18 cleanup step, affinity enrichment by direct click reaction to functionalized, magnetic beads is performed (3). This allows for stringent washing followed by acidic hydrolysis for elution. Subsequently, size exclusion chromatography (SEC) is applied to separate crosslinked from monolinked peptides (4). The resulting fractions are subjected to LC-MS/MS analysis (5). This graph was created in https://BioRender.com.

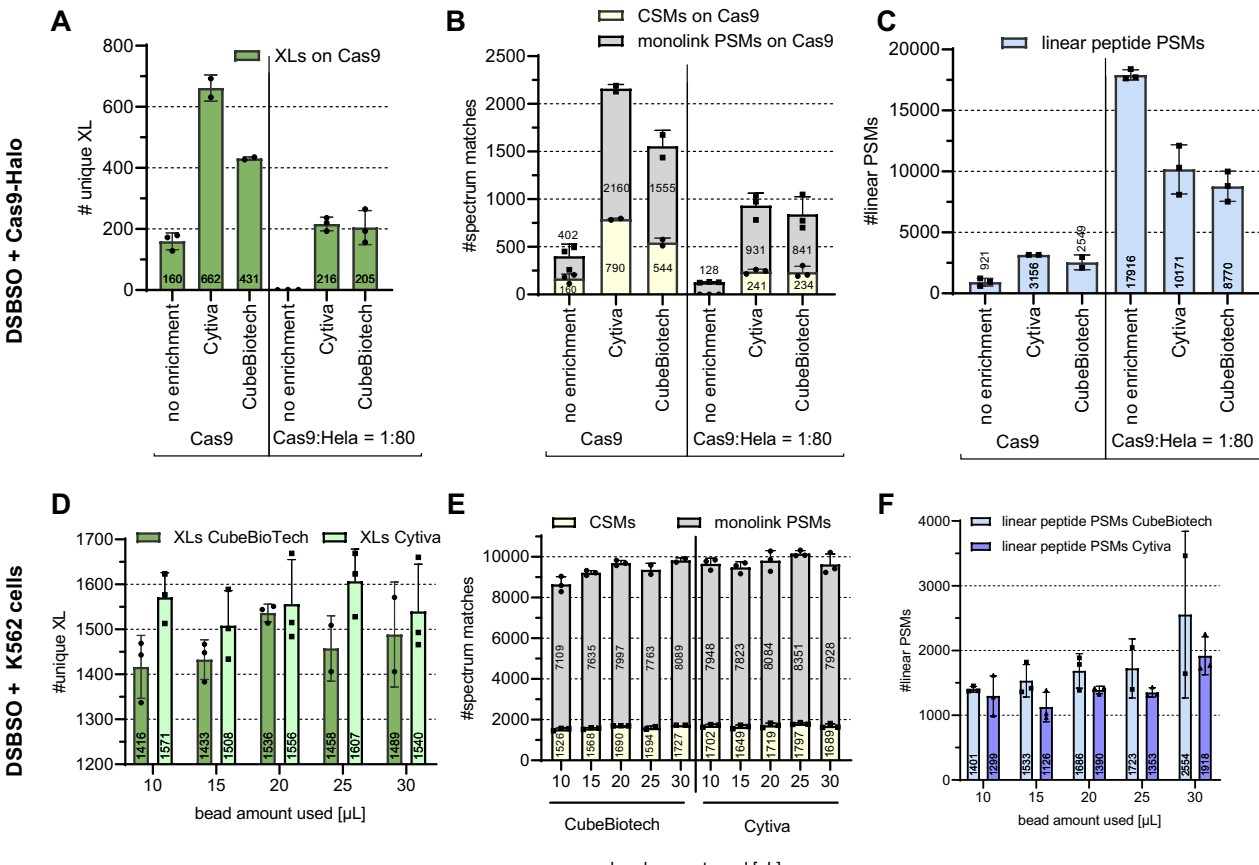

**Fig. 2 | Benchmarking magnetic beads from different vendors and finding optimal enrichment conditions.** A–C 20 µg DSBSO crosslinked Cas9-Halo peptides, spiked into a non-crosslinked background from human HeLa peptides (1:80, w:w), were either enriched using magnetic beads from Cytiva or Cube Biotech as indicated using 20 µL bead slurry and using the exact same processing workflow. Data was searched against a database containing 20,585 proteins (Cas9 + human proteome). D–F Living K562 cells were crosslinked, and enrichment was applied using both bead types under variation of the used bead slurry volume as indicated.

Data was searched against the human proteome. Black dots indicate values of individual replicates and bars indicate identified average numbers of unique XL residue pairs (**A**, **D**), CSMs and monolink PSMs, carrying a DSBSO modification (**B**, **E**) or PSMs from linear peptides without any DSBSO modification (**C**, **F**) at 1% FDR level, and error bars indicate their standard deviation, n = ≥2 (**A**–**C**) and n 3 (**D**–**F**). Peptides were separated using a 2 h gradient in trap-and-elute configuration, and data were recorded on an Orbitrap Eclipse instrument with detailed settings indicated in the "Methods" section.

Biotech clumped together in the absence of any detergent in some cases. The Cytiva beads did not clump and precipitated fast and completely on the magnet (estimated by visual inspection), also in the absence of detergent.

Next, we extended our tests to in vivo crosslinked K562 cell samples corresponding to 0.25 mg protein content. We titrated the amount of bead slurry used for enrichment and found a surprisingly small correlation between the applied bead amount on the resulting CSMs/unique crosslink sites (Fig. 2D–E). Additionally, only a minor increase in non-specific binders upon increasing the excess of bead material (Fig. 2F) was observed. Given the better properties concerning handling, slightly improved XL numbers, and slightly reduced background of linear peptides, we chose to proceed with the Cytiva beads for all subsequent studies. We identified 25 µL bead volume as the optimal condition between having enough binding groups and offering a higher surface for unspecific binding. Of note, this correlates to a bead volume/protein ratio of 100 µL/mg protein, and this ratio was also kept for the subsequent upscaled studies.

**Proof of principle study in K562 cells**

We applied the now refined affinity enrichment settings to the full workflow for in vivo DSBSO crosslinking adopted from the work of the Huang lab[9] (b). To streamline the workflow and potentially reduce sample losses on the (long) way from the cell culture to the mass analyzer, we aimed to further streamline the workflow by reducing the number of processing steps.

Omitting C18 cleanup ahead of the affinity enrichment step led to only slightly reduced identifications of crosslinks by 13% (Fig. 3A, B) while this unexpectedly helped to reduce the background from linear peptides by ~38% on average (Fig. 3C). We hypothesize that the presence of 0.5 M NaCl (added in the elution buffer of FASP) did not adversely affect the robustness of the click reaction, while it lowered unspecific binding of peptides to the bead-surface. When skipping the C18 cleanup step and applying size exclusion chromatography (SEC) as an orthogonal enrichment strategy instead, overall workflow sensitivity was dramatically improved leading to >3000 unique crosslinked sites identified on average from K562 cells or nuclear extracts from K562 cells and within four SEC fractions measured on the LC-MS (Fig. 3D). This improved sensitivity seems to mainly originate from reduced sample complexity with a lower background of linear peptides (Fig. 3F). In contrast, the obtained average ratio of CSMs/ monolinked peptides initially appears to remain at ~0.2–0.25, independent of SEC enrichment (Fig. 3B, E). However, a sufficient separation of monolinked from crosslinked peptides was facilitated via SEC, specifically in the early fractions, containing more and larger crosslinked peptides. In these early fractions, the CSM/monolink ratio was improved to >6 and drops to <0.2 for later eluting fractions (Supplementary Fig. 2). Of note, a complete separation of monolinked from crosslinked peptides was not expected, and our results successfully reproduce previous reports for SEC-based crosslink enrichment[12]. In this context, we noticed that the vast majority of crosslink

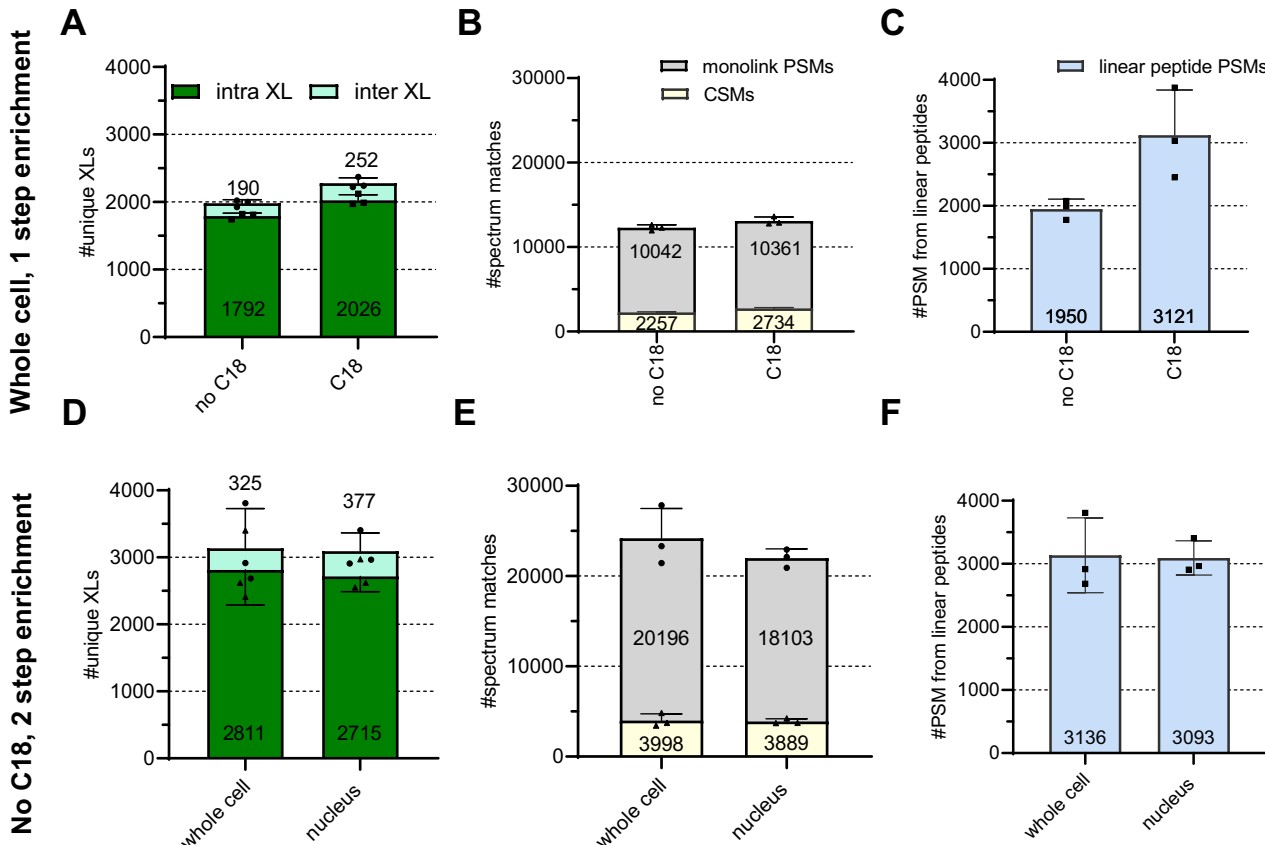

**Fig. 3 | Proof of principle study—DSBSO in vivo. A–C** Living K562 cells were DSBSO crosslinked and processed with or without a C18 cleanup step ahead of affinity enrichment using the Cytiva beads, but without an additional enrichment. **D–F** Whole K562 cells or their nuclear extracts were DSBSO crosslinked employing a streamlined workflow without C18 enrichment but including orthogonal SEC fractionation. Bars indicate identified average numbers of unique inter- and intra-protein XL residue pairs after combined analysis of four SEC fractions (**A, D**), CSMs, and monolink PSMs, carrying a DSBSO modification (**B, E**) or PSMs from linear peptides without any DSBSO modification (**C, F**) at 1% FDR level. Bars show averages, dots show values from each replicate, and error bars indicate their standard deviation, *n* = 3 independent replicates.

IDs, namely ~87% and ~90% when crosslinking the entire cell or the enriched nuclei, respectively, were found in only one SEC fraction (Fig. 4A, B). To select the best-performing fraction, measurement of all fractions of potential interest is needed at least for one replicate, though. For repetitions and experiments with similar settings, experience from that initial measurement can be used to make a fast decision on fractions of interest for measurement. For initial selection of fractions to be subjected to LC-MS, we used the obtained UV trace from SEC and selected early fractions starting at the point where the UV signal started to rise and ending where a huge background peak appears (Supplementary Fig. 3).

In conclusion, although fractionation was performed in this workflow, we propose that only a single fraction suffices for comparable coverage. As MS acquisition time is often limited and expensive, this strategy demonstrates another way to streamline this workflow, especially for larger sample cohorts in biological/medical studies.

To achieve maximal coverage of crosslink matches from our data, we decided to run a shared analysis of files from all fractions towards databases of different sizes (Fig. 4C, D). The assumption that too large databases might cause problems (next to extended search times) is further based on the number of redundant sequences within the database, increases with increasing size. This leads to ambiguous crosslink IDs as described earlier by the Bruce lab[13].

To generate reasonable databases of smaller size, we first generated a shotgun database from *n* = 5 biological replicates of non-crosslinked K562 samples, yielding 5659 proteins. From that, we created smaller databases containing only sequences from proteins of which at least 10 or 50 peptides

were found in the shotgun data, yielding 2318 and 91 remaining proteins, respectively. 19 and 20 unique peptides were found from the main proteins of interest in this study, DDX39A and its homolog DDX39B, but both proteins were kept in the database even for the most stringently filtered database requiring 50 peptides/protein. With this, we aimed to maximize crosslinks potentially detectable on those two proteins. As shown in Fig. 4C, D, using the full shotgun database was most advantageous in both systems, whole cells and nuclear extracts, to maximize overall crosslink numbers. We hypothesize that this is because of less possible random (decoy) hits when using a reduced database size, resulting in a less stringent score cutoff applied (dropping by ~21% and 28% for whole cell and nucleus samples, respectively, see Fig. 4C, D) while maintaining a 1% FDR threshold. We further checked for changes in score distribution on the CSM level and found that a slight shift towards lower scores was accepted for 1% FDR with databases smaller than the full human proteome (i.e., mode score drops from ~300 to 250, Supplementary Fig. 4A, B). The relative fraction of decoy-decoy + decoy-target + target-decoy (named decoy for simplicity here) vs. target-target (named target) hits remains at a comparable level, though (Supplementary Fig. 4C), and after FDR filtering, 1% of decoy hits remain as expected (Supplementary Fig. 4D). We manually checked the spectrum quality of the lowest scored spectrum accepted at 1% FDR and of representative spectra close to the mode score from the largest (20584 sequences) and smallest database search (91 sequences) done, with no significant differences in quality becoming obvious (Supplementary Fig. 5). This is in line with our expectation from an earlier study, where we used a peptide library to experimentally validate FDR[14]. There, we found that reducing the

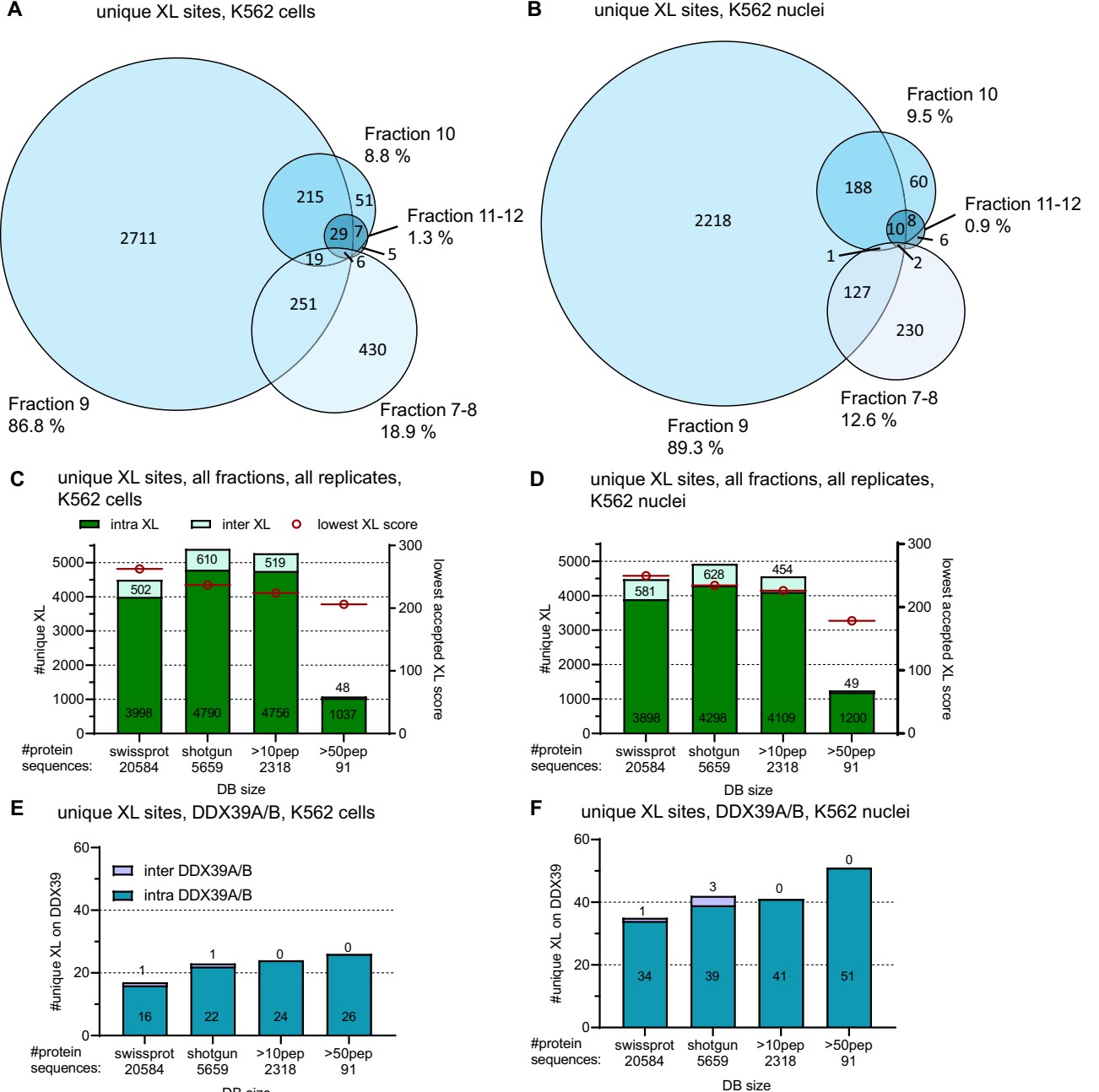

**Fig. 4 | Improving coverage with minimal effort: commonly identified crosslinks across SEC fractions and influence of database search on overall coverage.** Venn diagrams show commonly identified unique XL sites across the acquired SEC fractions from a representative replicate each from whole K562 cells (**A**) and K562 nuclei (**B**), respectively. **C–F** Changes of crosslink ID numbers depending on database size. Bar plots showing either all XL unique XL sites from whole cells (**C**) and nucleus (**D**) or within those only unique XL sites found on either DDX39A or B from whole cell (**E**) or nuclear extract (**F**) samples. All fractions and $n = 3$ replicates for each condition were searched together at 1% FDR.

database size in a similar range and using the same search engine even reduced the experimentally validated FDR in samples with or without enrichment performed. This indicates that in the database size range applied also in this study, enough decoy hits are still found to perform a proper target-decoy based FDR validation.

For DDX39, reducing the database size led to slightly higher intra-link numbers for DDX39 proteins (Fig. 4E, F) while lowering overall numbers, likely due to crosslinked proteins being removed from the database. No additional inter-protein hits for DDX39 were found using databases smaller than the 5659-protein-sized shotgun database. Simultaneously, the risk of less confident intra-links being reported as false positives is higher with very small databases. Consequently, we decided to continue with data from the full shotgun database for further data processing.

## Study of DDX39B

The ATPase DDX39 is a central molecular switch that directs mature mRNA ribonucleoprotein complexes and thereby controls mRNA export[15]. To validate the expected primary localization of DDX39B in the nucleus within the herein used cell-based system, we made use of a clone expressing the GFP-tagged protein version. DNA was counterstained using DAPI, confirming co-localization with DDX39B as expected (Fig. 5A). In conclusion, we decided to focus on samples measured from nuclear extracts

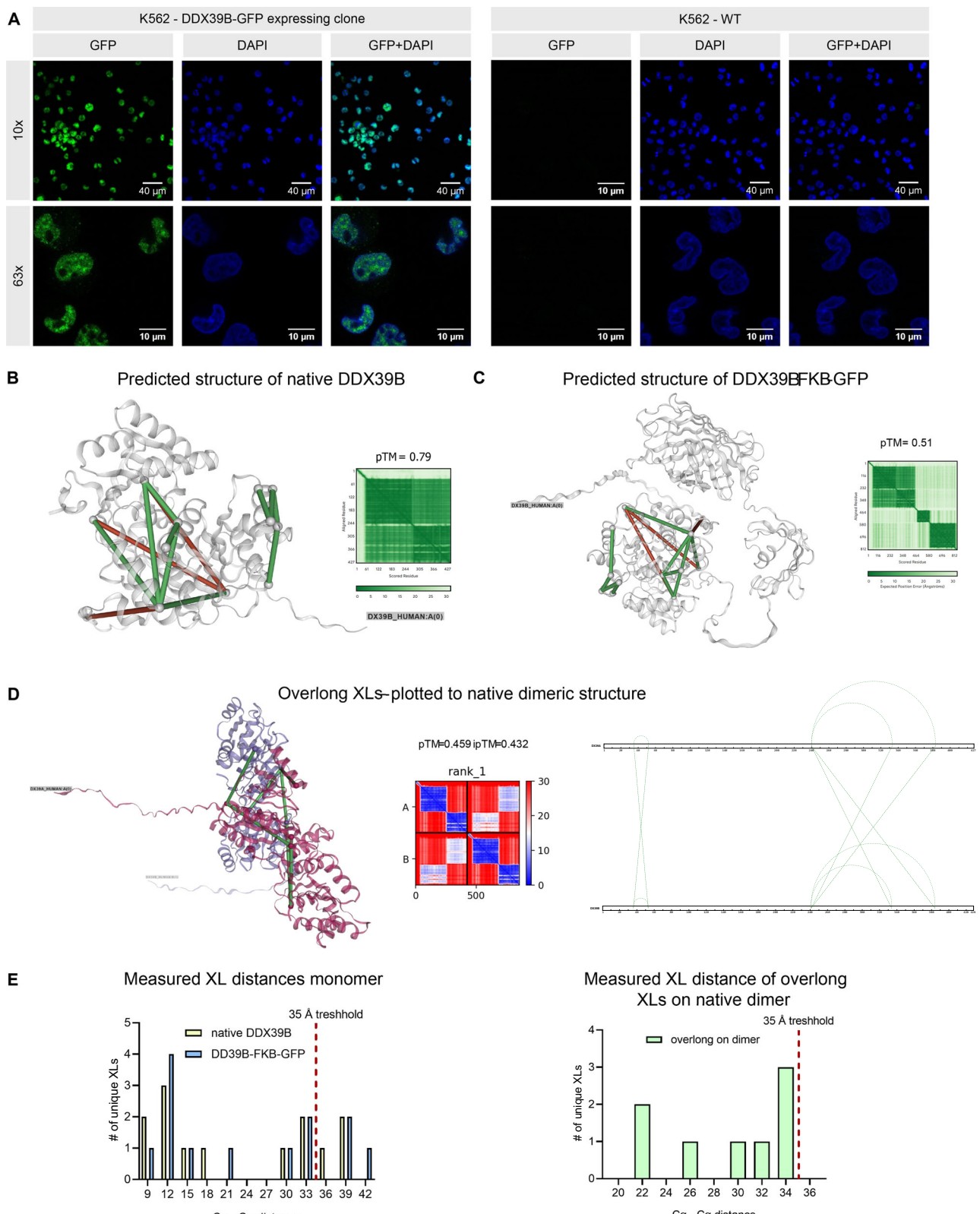

**Fig. 5 | Localization and structure prediction of DDX39. A** DDX39B-GFP expressing and wt-K562 cells were DNA stained using DAPI and inspected by fluorescence microscopy at 10x or 63x magnification as indicated. GFP signal shown in green and DAPI signal shown in blue. Crosslinks found within the nuclear extracts/ analyzed using the shotgun database, plotted on the native DDX39B (**B**) or the tagged DDX39B-FKB-GFP (**C**) rank 1 3D structure model predicted from AlphaFold3 and their respective diagnostic plots. For ambiguous crosslinks, only the shortest possible connection, when plotted to the predicted 3D structure from AlphaFold2, is shown. Shown in green are crosslinks ≤35 Å and in red links >35 Å. **D** Possible K-K connections with ≤35 Å from previously non-satisfied crosslinks after plotting onto the monomeric DDX39B structure, now plotted to the predicted structure of the DDX39A and B complex and allowing links to fit on either protein or across both proteins. **E** Histogram of measured Cα to Cα distances when plotting found crosslinks to the predicted 3D structures as indicated.

where higher concentrations of DDX39 are expected (as confirmed by yielding higher XL-ID numbers, shown in Fig. 3C, D).

Of those 39 identified intra-protein crosslinks for DDX39A or B, most could not unambiguously be annotated to only one homologous protein sequence (see Fig. 5A, with unambiguous links shown as dashed lines). The sequence identity of DDX39A and B is 89.95% when using BLAST®[16] to compare both sequences. We therefore checked for proof that DDX39A and B are present and crosslinked in the sample: We found peptides of 6 ambiguous sequences connected 383 CSM (where peptide A and B of the crosslink are ambiguous). For DDX39A peptides from 8 unambiguous sequences connected 223 CSM were found (where at least one of the connected peptides is unambiguous). For DDX39B peptides from 6 unambiguous sequences were found connected to 206 CSM (where at least one of the connected peptides is unambiguous). We further found an unambiguous crosslink from DDX39A and B to CHTOP, respectively. We, however, found exclusively ambiguous crosslinks from DDX39A to DDX39B (see below; Fig. 5D). The covered ambiguous and unambiguous sequence is shown within Supplementary Fig. 6. All found links can be found within Supplementary Data 7.

We first focused on DDX39B and predicted its structure with or without the attached FKB-GFP tag using AlphaFold3[17]. To remove ambiguous hits, we further selected only the shortest possible crosslink distance as measured on this predicted structure for plotting and further processing (Fig. 5B, C). We decided on applying a distance cutoff of 35 Å as suggested by Jiao et al.[9] leading to ~77% of unique DSBSO links in agreement with the predicted structure. The number of overlong crosslinks was not altered in dependence on the FKB-GFP tag included for prediction. Of note, the predicted template modeling score is lowered for the tagged prediction vs the native from 0.79 to 0.51 respectively and the resulting average crosslink length was slightly elongated in presence of the tag (Fig. 5F). All distance measurements were performed using the respective rank 1 models (Fig. 5). Notably, since DDX39A and DDX39B are highly homologous, all overlength crosslinks found are ambiguous and might relate to an intra-protein connection on DDX39A or to an inter-protein connection across DDX39A and B in dimeric form. We predicted the structure of the respective dimer (Fig. 5D). Our experimental data suggests that the predicted dimeric structure could be indeed valid with crosslink connections satisfying the maximum distance threshold for DSBSO for all previously overlong links (Fig. 5D). We therefore assume a monomeric and dimeric protein form might be present within the nuclear extracts both captured in our data. Of note, our findings are in line with previous reports describing that DDX39A can heterodimerize with DDX39B, which would inhibit its function[18]. In that previous study, DDX39A was overexpressed, though, which was not the case in this study. The here used CRISPR tagged cell line excludes the possibility of overexpression and data from western blot controls suggests that tagging makes the protein a bit less stable, resulting in slightly reduced protein levels compared to the wild-type (Supplementary Fig. 7). In conclusion we assume that the found crosslinks cannot be explained by complexes formed artificially due to overexpression. No other study is known to the authors that would investigate DDX39A/B complexes as standalone within their native environment; however, Pühringer et al.[19] reported 4 UAP56/DDX39 entities in complex within the essential transcription–export complex. In their structure, the UAP56/DDX39 molecules are positioned very close to each other, with the distance between the C-terminal RecA lobe of one UAP56 and the N-terminal RecA lobe of the neighboring one within a 20 Å range. This could explain crosslinks within this complex in general, although the exact link position found within this study is not covered by that structure (PDB: 7APK). In addition, a recent study from Hohmann et al.[15] suggests that once UAP56 is bound to RNA, it dissociates from THO, resulting in some flexibility in how neighboring UAP56 molecules are positioned once released. To sum up, there is no published hard evidence for UAP56 dimerization, but there is evidence for DDX39A/B molecules being in very close proximity in vivo, which would enable crosslink formation.

While this study focuses on method development, drawing solid biological conclusions from our crosslink data alone is not possible and would require additional confirmation by orthogonal experiments. Nevertheless, the current data showcase how hints for the presence of dimeric forms can be found by combining crosslinking data with structure prediction data.

In addition to these DDX39 intra-complex connections, we detected crosslinks to CHTOP (Fig. 6A). We compared our findings to known PPI for DDX39 using the STRING database[20] (Fig. 6B) and found that a physical interaction of DDX39A and B and its interaction with CHTOP are known, which we confirmed in this study. Our data demonstrates an interaction of both DDX39A and B to K226 of CHTOP, which confirms earlier results by Fasci et al.[21], who identified this exact crosslink connection in nuclear extracts from U2OS cells using a different crosslinker (DSSO).

To validate that our data are consistent with structural models, we further predicted the structure of DDX39A and B in complex with CHTOP, respectively (Fig. 6C, D) using AlphaFold2[22]. We found that the unambiguous inter-protein connection from DDX39A-CHTOP (16.1 Å) and DDX39B to CHTOP (16.1 Å) well aligns with the expected maximum distance threshold. No intra-protein crosslink was found on CHTOP, and only a single unambiguous intra-protein link was found on DDX39B, which also aligns with the expected distance. However, for DDX39A, 4 out of 10 unambiguous intra-protein crosslinks are overlength. Hence, we tried to use the XL data to guide structure prediction with the help of AlphaLink2[23–25]. This resulted in an altered structure with only minor changes of the crosslink lengths, and no change in the fraction of crosslinks accepted within the used 35 Å threshold (Fig. 6C, D). We hypothesize that those intra-protein links might not co-exist with the CHTOP connection within the same protein-complex molecule but rather reflect different complexes present in parallel in solution (similar as shown for Fig. 5B–D, where some links fit to dimeric, others to monomeric structures).

## A comprehensive PPI network

In addition to the data generated on DDX39, we obtained a comprehensive PPI network from the K562 model system. A detailed list of all crosslink matches can be found in Supplementary Data 5 and 6 for results searched towards the shotgun database from whole cells and nuclear extracts, respectively. Identified crosslinked proteins were distributed across several cellular compartments with the vast majority found within the cytosol or nucleus (Fig. 7A). When crosslinking within nuclear extracts, ~80% of all crosslink matches could be annotated to the nucleus or had an ambiguous annotation with pa possible localization in the nucleus, which enables purity estimation of our samples after nuclei enrichment. Proteins with ambiguous localization annotation, that are present within the nucleus and the cytosol according to their GO annotations, are depicted as such in Fig. 7A and were counted as co-localization. As little as 0.5% and 0.2% for whole cell and nuclear extract, respectively, of all crosslinks were between proteins from different cellular compartments. Such crosslinks theoretically should not form. Although, of course, no proof for FDR, the prevalence of non-co-localizing crosslinks way below 1% indicates a potentially functional FDR control, while a higher prevalence would be concerning. Within the whole cell dataset, analyzed towards a generated shotgun database (see Fig. 4C), we found a total of 311 PPI from non-ambiguous crosslinks. Gene enrichment for Gene Ontologies of biological processes, cellular compartments, and molecular function was performed. We found networks within the ribosomal complexes, chaperonin-containing T complex, the spliceosome, and proteasome, and others (Supplementary Fig. 9, Supplementary Data 5 for all crosslinks found and Supplementary Data 8 for novel interactors not listed in the String database).

Within the nuclear extract dataset, we found a total of 333 PPIs from non-ambiguous crosslinks. We again performed gene enrichment and found networks within the spliceosome, DNA replication, chromatin compartments, mRNA export proteins, including DDX39B and others (Fig. 7B, Supplementary Data 6 for a list of all crosslink connections found). Our data thereby provide valuable insights into molecular pathways within the cell and include 56 (14% of all PPI) potential novel interaction partners,

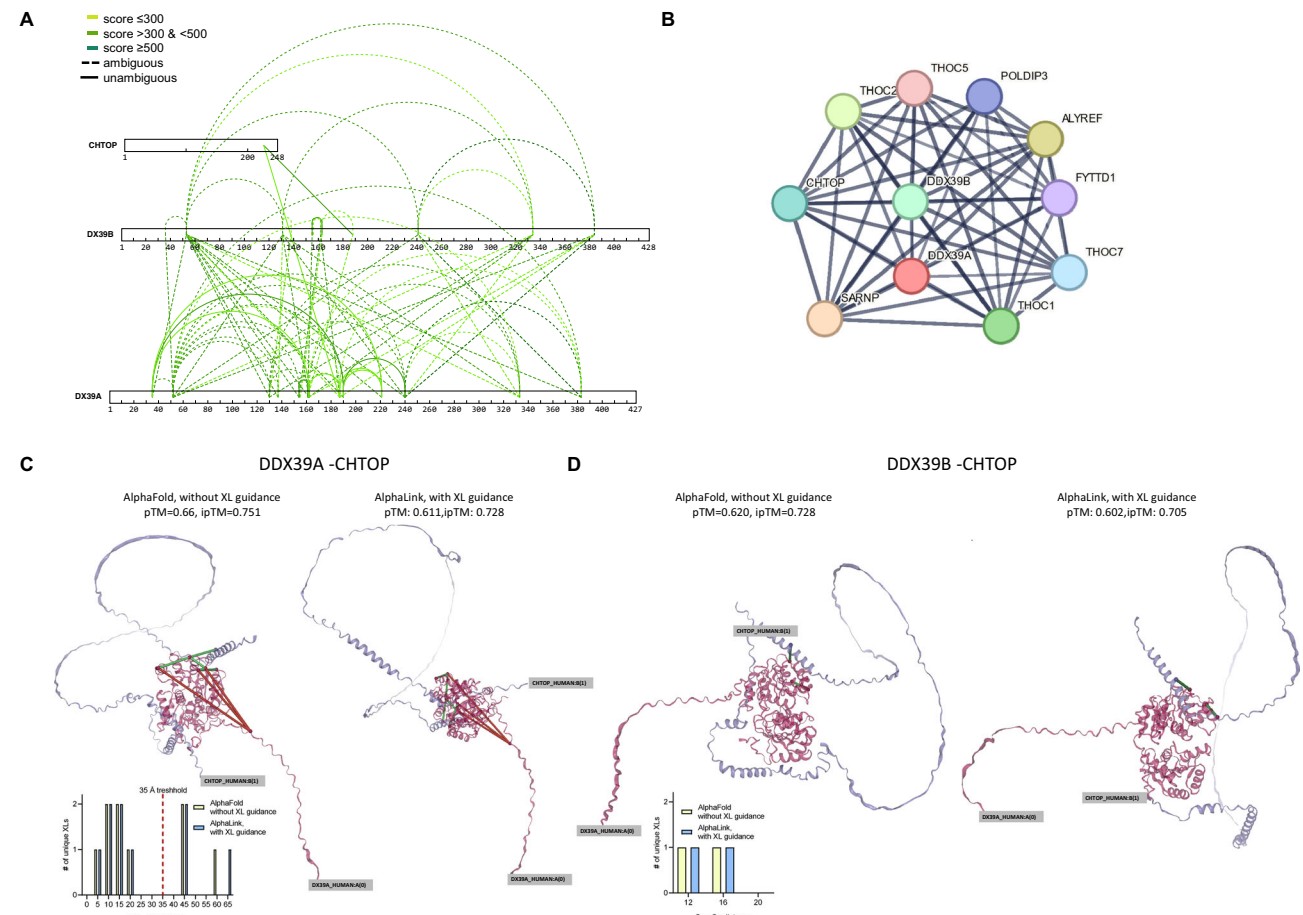

**Fig. 6 | Identified PPI of DDX39A/B. A**: All XLs found on DDX39B (shown without tag) and its direct interaction partners. Dashed lines indicate ambiguous links which might originate from intra-protein links or inter-protein links to the homologous sequence of DDX39A/B respectively. Solid lines indicate unambiguous XLs and darker green indicates higher confidence. **B** Known interaction partners of the DDX39A/B complex as given in the STRING database. Panel **C** and **D** show unambiguous crosslinks (shown as solid line in A) plotted to the predicted Alphafold2 or Alphalink structures of DDX39A (panel **C**) or DDX39B (panel **D**) to CHTOP. Structures were predicted without using XL data or using the plotted unambiguous links for guidance respectively. The histograms show the resulting measured Cα to Cα crosslink-distance distribution.

not listed in the String database (Supplementary Data 7). In line with the findings suggested by Fürsch et al.[26], our data reports crosslinks primarily on more abundant proteins within the cell (Supplementary Fig. 8). However, our crosslink data covers ~5–6 orders of magnitude in abundance of crosslinked proteins showcasing that at least some crosslinks even from low abundant proteins have been detected thanks to the effective enrichment strategy. Moreover, within the nuclear extract, we were able to cover more low-abundant proteins compared to the whole cell crosslinking approach (Supplementary Fig. 8).

## Methods

Figure 1 provides an overview of the workflow, with detailed explanations given in the following sections.

### Reagents

Purified recombinant Cas9 from *S. pyogenes* fused with a Halo-tag was generated in-house, as described by Deng et al.[27]. Azide-A-DSBSO was obtained from Sigma-Aldrich (#909629) or from Hycultec (only for experiments of Fig. 2A–C, #HY-157414). For enrichment bead generation, dibenzocyclooctyne-amine (DBCO-amine, #761540, Sigma-Aldrich) was coupled to NHS Act Sepharose® 4 Fast Flow (Cytiva, #GE17-0906-01), NHS Mag Sepharose® (Cytiva, #28951380) beads, or Pure Cube NHS act. MagBeads (Cube Biotech, #50405) as indicated. The prepared beads were stored as a 50% slurry in a 1:1 ethanol:water mixture. Trypsin gold was purchased from Promega (Mannheim, Germany), and lysyl

endopeptidase (LysC) was from Wako (Neuss, Germany). Benzonase—pharmaceutical production purity—was purchased from Merck (Darmstadt, Germany). All other solvents were purchased from Fisher Chemicals if not otherwise indicated.

### Cell culture and nuclear extract preparation

DDX39B-dTag-GFP mutant K562 (DSMZ) cells were generated as follows: The DDX39B locus was CRISPR modified with a C-terminal FKBP-GFP tag using the Alt-R system of IDT, including gRNA, electroporation, and HDR enhancers. A repair template containing the tag sequence was generated by PCR with ~50 bp long homology arms. The gRNA target site was: TCTTCTCAGTTGAACAGACA, the homology arm sequences: 5′HA TTTCATGCCTTATTCTGACCATGCTACGTTTTCTTCTCAGTTGA ACAGACACGA and 3HA TGTCCTCTCCTGAAGGACAGACGGTCA CATTCCAAAATGGGCGAGTCTTTTA. Cas9 ribonucleoprotein complexes and template DNA were electroporated with a MaxCyte STx electroporator. Cells with high GFP signal were FACS sorted two times consecutively, single cell clones with homogenous GFP signal were tested by genomic PCR for correct tag insertion and western blotting for correct fusion protein size.

K562 wild-type (WT) and DDX39B-dTag-GFP mutant cells were grown in RPMI 1640 medium, supplemented with 10% fetal bovine serum (FBS), 1 mM sodium pyruvate, 1% penicillin/streptomycin, and 2% L-glutamine. The cells were cultured at 37 °C in a humidified incubator with 5% $CO_2$. Once the cultures reached the appropriate density, the cells were

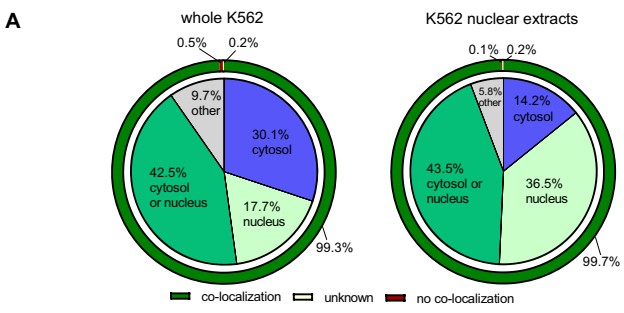

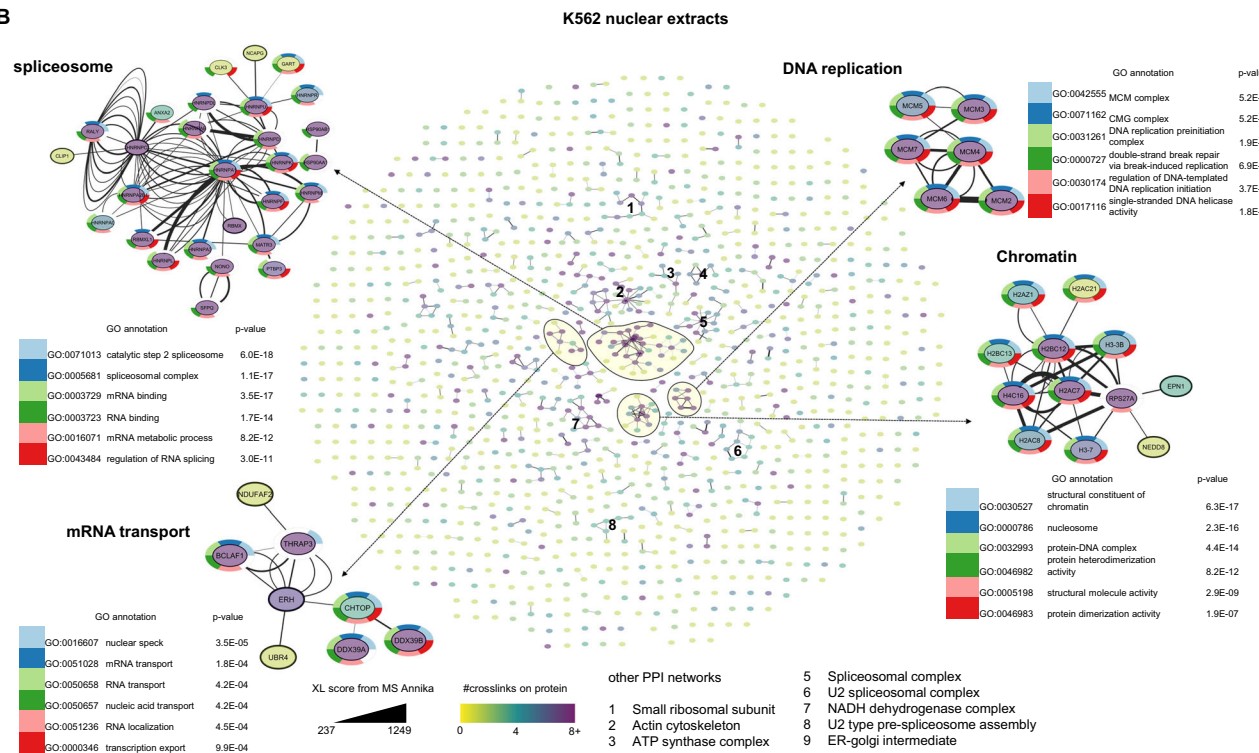

**Fig. 7 | PPI found in vivo XL within K562 cells. A** Distribution of cellular compartments of crosslinked proteins after crosslinking intact cells or their nuclear extracts, as indicated. Data based on the GO annotations of DSBSO connected proteins. The outer sphere shows the fraction of connected proteins within the same cellular compartment (co-localization) or within different compartments. **B** PPI network as found from nuclear extract crosslinking samples after analysis against a shotgun database (see Fig. 3C) with correlated groups annotated. In total, 333 non-ambiguous PPIs from 628 heteromeric crosslinks were found. Selected sub-PPI-networks are highlighted with unambiguous inter-protein crosslinks shown. The edge thickness indicates the best MS Annika CSM score for each unique crosslink shown, found within a 1% FDR threshold, with high scores indicating higher confidence. Node colors indicate the total number (inter- and intra-protein) of crosslinks found on each protein. The connected gene names are annotated for each protein, and the top-6 gene annotations after gene enrichment are shown.

washed with phosphate-buffered saline (PBS), divided into aliquots, and stored in liquid nitrogen for future use.

To isolate nuclei, a modified Rapid, Efficient, and Practical[28] protocol was used. The cells were lysed gently in ice-cold PBS containing 0.1% Igepal CA-630. Nuclei were collected through centrifugation and washed to remove any remaining cytoplasmic components. The purified nuclei were aliquoted and stored at −80 °C for further experiments.

### XL reaction Cas9
Twenty micrograms of purified recombinant Cas9 from S. pyogenes fused with a Halo-tag at 1 mg/mL in 200 mM HEPES (pH 7.6) was crosslinked by adding a 4 mM DSBSO in dry DMSO stock to a final concentration of 0.2 mM DSBSO. The crosslinking reaction was incubated at 37 °C on a thermoshaker set to 1000 rpm for 1 h. Crosslinked Cas9 samples for initial check-experiments were either used without additional enrichment or spiked in a 1:80 (w/w, 20 μg Cas9 + 1.6 mg HeLa) ratio into non-crosslinked Pierce™ HeLa Protein Digest (Thermo Fisher), followed by DBCO bead-based enrichment using 30 μL bead

slurry and as described within the section "Affinity enrichment of XL-peptides".

### Chemical crosslinking of K562 cells and nuclei
2E8 frozen cells or nuclei from the same number of cells were thawed quickly by resuspension within a crosslinking buffer consisting of 20 mM HEPES (pH 7.6), 150 mM NaCl, and 1.5 mM MgCl₂. Crosslinking was initiated immediately after resuspension by directly adding the DSBSO powder to the cells/nuclear extracts to reach a final concentration of 2 mM DSBSO. Samples were incubated while shaking gently at 37 °C for 1 h at 1000 rpm. One microliter of Benzonase was added for the final 30 min during the crosslinking reaction to facilitate DNA digestion. No quenching of crosslinking was performed as potential excess and reactive linker was removed by FASP thereafter.

### Protein quantification and filter-aided sample preparation (FASP)
The samples were lysed using a Bioruptor sonicator set to 5-min cycles (30 s on and 30 s off) at 4 °C. After lysis, the samples were clarified by

centrifugation at $14,000 \times g$ for 15 min at 4 °C and kept on ice until further processing.

Protein concentrations were measured using the Bio-Rad Protein Assay Dye Reagent (Cat. No. 5000006), and the obtained lysate was aliquoted into portions of 1 mg at a harmonized concentration of 5 mg/mL. One milligram of protein was used for each replicate for subsequent processing: The sample was divided into three FASP filters (0.33 mg per filter; Merck Millipore, Cat. No. MRCF0R030). The peptides eluted from these filters were again pooled into a single tube per replicate.

Filters were equilibrated with 8 M urea in 0.1 M Tris-HCl (pH 8.2) by centrifugation at $14,000 \times g$ for 15 min. Proteins were loaded onto the filters in the same buffer and concentrated by centrifugation, followed by a wash step with equilibration buffer. Reduction was carried out using 2 mM Tris(2-carboxyethyl)phosphine (TCEP) in 8 M urea, and alkylation was performed with 10 mM iodoacetamide (IAA) in 8 M urea, both at room temperature. After this step, the workflow diverged: For the general FASP approach, filters were washed twice with 100 mM HEPES (pH 7.3) to remove residual reagents, while for the modified FASP with C18 cleanup, an alternative washing protocol was applied, as described in the respective section.

Proteins were digested sequentially, starting with lysyl endopeptidase (LysC) at a 1:100 (w/w) enzyme-to-protein ratio in 100 mM HEPES (pH 7.3) at 37 °C overnight, followed by trypsin digestion under identical conditions for 4 h. Peptides were first eluted by centrifugation, followed by sequential elution with 100 mM HEPES (pH 7.3) and 0.5 M NaCl. The eluates from the three filters were pooled to generate a final peptide sample.

## Modified FASP with C18 cleanup
For the modified protocol, filters were first washed with 25 mM ammonium bicarbonate (ABC) buffer (pH 9.2) containing 8 M urea before digestion. LysC digestion was conducted in 5.45 M urea and 25 mM ABC buffer (pH 9.2), while trypsin digestion was performed in 1.5 M urea and 25 mM ABC buffer. The peptides were eluted as described and then further purified using Sep-Pak® tC18 cartridges (Waters, Cat. No. WAT054960).

Sep-Pak® tC18 cartridges were equilibrated with a sequence of methanol, 70% (v/v) acetonitrile (ACN) with 0.1% trifluoroacetic acid (TFA), and 0.1% (v/v) TFA. Acidified samples (adjusted to pH 3–4 with TFA) were loaded onto the cartridges. After washing with 0.1% (v/v) TFA, peptides were eluted using 70% (v/v) ACN containing 0.1% (v/v) TFA. To prevent DSBSO acetal hydrolysis, the eluates were neutralized with 1 M HEPES (pH 7.3). Finally, the samples were concentrated using a SpeedVac and reconstituted in 100 mM HEPES (pH 7.3) to ensure consistency with the conditions used in the standard FASP procedure.

## Preparation of DBCO-coupled beads for affinity enrichment
The NHS Mag Sepharose® beads manufactured by Cytiva were used for all in vivo XL studies. 500 µL of the bead slurry (~20% slurry) was washed immediately prior to usage once with 1 mL ice-cold 1 mM HCl. After that, beads were resuspended in a mixture of 385 µL 50 mM HEPES pH 7.5 and a stock solution of 64 µL 10 mg/mL Dibenzocyclooctyne-amine (761540-10MG, Sigma) in dry DMSO. The resulting suspension was incubated for 2 h at RT on a rotator, followed by 2 × washing using 50 mM HEPES pH 7.5, 1× washing with 1 M NaCl, and 2 × washing with 100 mM Tris-Cl pH 7.5. For storage at 4 °C, beads were resuspended in 100 mM Tris pH 7.5 to reach a final volume of 1000 µL.

## Affinity enrichment of XL-peptides
Beads for affinity enrichment were extensively washed to ensure optimal binding conditions. Initially, they were washed five times with 1 M NaCl in 50 mM HEPES (pH 7.3) to remove contaminants, followed by five additional washes with 50 mM HEPES (pH 7.3) for equilibration.

Samples were incubated with prepared beads under rotation for 2 h at room temperature or overnight at 4 °C, depending on the experiment. For bead optimization, ratios of 10, 15, 20, 25, and 30 µL of beads per 0.25 mg of protein were tested, with 25 µL per 0.25 mg found to be optimal. This

condition was later upscaled (4x) to 100 µL of beads per 1 mg of protein for all in vivo studies. Cas9 tests were performed with 20 µL of beads per 20 µg of recombinant Cas9 protein, with an additional 20 µg of HeLa digest for spiking experiments, mimicking cellular background peptides.

After incubation, the supernatant was removed using magnetic separation. Beads were washed sequentially with a volume five times greater than the bead volume. The washing protocol included five washes with 1 M NaCl in 50 mM HEPES (pH 7.3), followed by three washes with 10% (v/v) acetonitrile (ACN), and four washes with MilliQ $H_2O$ (0.2 µm filtered). To ensure thorough cleaning, the beads were transferred to a new tube for a final wash with MilliQ $H_2O$, minimizing any potential carryover before elution.

Crosslinked peptides were eluted in two steps using 2% (v/v) trifluoroacetic acid (TFA) in water. In the first step, two-thirds of the bead slurry volume was incubated for 1 h, followed by elution of the remaining one-third for 30 min. The eluates were combined, and those intended for immediate analysis were dried using a SpeedVac and resuspended in 0.015% DDM in 0.1% TFA. These samples were either resolved at 4 °C overnight or at room temperature for at least 30 min prior to analysis. For samples designated for SEC, the final concentration step adjusted the sample volume to ~45 µL to ensure compatibility with SEC workflows.

## Fractionation via SEC
SEC was performed using a TSKgel SuperSW2000 column (4 µm particle size, 4.6 mm ID × 30 cm) to separate peptides based on size. The column operated at a flow rate of 0.3 mL/min with detection at 214 nm. 0.1% (v/v) TFA in ddH$_2$O was used as loading buffer. The elution buffer consisted of isocratic elution with 30% (v/v) ACN and 70% (v/v) of 0.1% TFA in water. Samples were injected in their entirety, and fractions were collected at 1-min intervals.

Fractions were pooled based on the intensity of the elution peaks. Fractions with lower peptide intensity were combined (e.g., 7 + 8 and 11 + 12), while those with higher intensity (e.g., fractions 9 and 10) were processed separately. Prior to MS/MS analysis, all samples were resolved in 0.015% DDM and 0.1% TFA and injected at a final volume of 10 µL.

## Liquid chromatography–mass spectrometry (LC-MS) analysis
Analyses were performed using a Vanquish Neo UHPLC system (Thermo Fisher Scientific). If not given, an 800 ng sample was injected for each LC-MS run. Samples were loaded onto a PepMap Acclaim C18 trap column (5 mm × 300 µm, 5 µm particle size, 100 Å pore size; Thermo Fisher Scientific) in trap-and-elute mode, then switched in line with the analytical column for separation. For bead benchmark studies, separation was carried out on a PepMap Acclaim C18 analytical column (500 mm × 75 µm, 2 µm, 100 Å; Thermo Fisher Scientific) at a flow rate of 0.300 µL/min. All other analyses were conducted on an Aurora Ultimate 25 cm column (250 mm × 75 µm, 1.9 µm, 120 Å; IonOptiks) at 0.230 µL/min. The column temperature was maintained at 45 °C.

A linear gradient was applied, gradually increasing Buffer B from 2% to 45% over 120 min, followed by a transition to 95% Buffer B within a minute. The column was maintained at 95% Buffer B for 6 min before re-equilibration at 2% Buffer B. Buffer A consisted of 100% water with 0.1% (v/v) formic acid, while Buffer B was composed of 80% (v/v) acetonitrile with 0.1% (v/v) formic acid.

Mass spectrometric analyses were performed on an Orbitrap Eclipse Tribrid mass spectrometer (Thermo Fisher Scientific) equipped with a FAIMS Pro interface set to standard resolution, with compensation voltages of $-50 \pm 10$ V. The instrument was operated in positive-ion mode under a data-dependent acquisition Top 10 method. MS1 scans were acquired over an $m/z$ range of 375–1500 at a resolution setting of 120,000, with an automatic gain control (AGC) target of $1.2 \times 10^6$ (300% normalized). MS2 scans were generated using HCD at stepped collision energies of ~$27 \pm 6\%$, with a 1.2 $m/z$ isolation window, a resolution of 30,000, and an AGC target of $5.0 \times 10^4$ (100% normalized). Precursor ions were restricted to charge states 3–8, required a minimum intensity threshold of $2.5 \times 10^4$, and were

dynamically excluded for 30 min after one fragmentation event. The spray voltage was set to 2.3 kV, and the capillary temperature was maintained at 275 °C.

### Fluorescence microscopy

All imaging was performed on a Zeiss LSM 780 confocal microscope in laser scanning confocal mode. High-magnification images for nuclear co-localization were acquired using a Plan-Apochromat 63×/1.40 Oil DIC objective, while overview images were taken with a Plan-Apochromat 10×/0.3 M27 objective. EGFP (DDX39B-GFP) fluorescence was excited at 488 nm and detected at ~517 nm, and DAPI was excited at 405 nm with emission collected near 451 nm. The pinhole was set to roughly 1 Airy unit for both channels, and laser power attenuation ranged from about 96–99%. Detector gains were typically set between 822–852 for EGFP and 707–757 for DAPI, with digital gains of 3.86 × (EGFP) and 3.19 × (DAPI). Z-stacks of seven slices were acquired per field of view to capture nuclear volumes. Image processing was done via ImageJ Version 1.54 m.

### Immunoblotting analysis

Cell lysates of K562 wt or clone cells were subjected to SDS-PAGE followed by western blotting to PVDF membranes. Used antibodies were Abcam ab181059 against UAP56 and Abcam ab1791e against Histone H3.

### Data analysis

LC-MS/MS raw files were analyzed within Proteome Discoverer 3.1 (Thermo Fisher Scientific), using MS Amanda 3.0[29] for searching linear peptides and MS Annika 3.0[30] to search crosslinked peptides. Relative, label-free quantification was done using apQuant[31]. Results were filtered to 1% FDR at peptide, protein, CSM, and unique XL site levels. A DSBSO linker-based workflow, as recommended in the most recent MS Annika publication[30] (download: https://github.com/hgb-bin-proteomics/MSAnnika/raw/master/workflows/PD3.0/DSBSO_MS2.pdAnalysis), was used for data analysis. Searches were performed on databases of different sizes, as indicated for each result within this study individually.

### Post processing

For data visualization and statistics, GraphPad Prism 8.0 (GraphPad Software Inc.) was used. Venn diagrams were plotted using DeepVenn[32]. Post processing to visualize 3D models was performed using xiVIEW[33], for proteome-wide interaction network visualization and gene enrichment analysis, Cytoscape v3.10.3[34] was used. Results obtained from Annika in Proteome Discoverer were exported to xiview using a custom Python script available on GitHub (https://github.com/hgb-bin-proteomics/MSAnnika_exporters/blob/develop/xiViewExporter_msannika.py). 3D structure prediction was performed using AlphaFold3[17] on a local cluster. AlphaFold[22] on a Google Colab sheet[25] was used to predict the dimeric DDX39A-B structure, or using DDX39A/B—CHTOP structures using the suggested default settings (https://colab.research.google.com/github/sokrypton/ColabFold/blob/main/AlphaFold2.ipynb?pli=1#scrollTo=R_AH6JSXaeb2). DDX39A/B—CHTOP structures were further predicted, including crosslink data from unambigous links found within this study, using AlphaLink2[23–25] on a Google Colab sheet (https://colab.research.google.com/github/Rappsilber-Laboratory/AlphaLink2/blob/main/notebooks/alphalink2.ipynb#scrollTo=j-xTD0QubEN-).

### Reporting summary

Further information on research design is available in the Nature Portfolio Reporting Summary linked to this article.

## Discussion

Despite tremendous improvements already made in the field of crosslinking mass spectrometry, with a multitude of different linker chemistries and sample preparation protocols being developed, the generation of comprehensive in vivo crosslinking data is still challenging. Within this study, we provide an easy-to-use, streamlined workflow for this purpose. We provide guidance on which magnetic bead type is best suited, and at what excess over DSBSO crosslinker enrichment beads perform optimally. By removing a C18-based purification step, we reduce potential sample losses without suffering from relevant final crosslink coverage. This is especially advantageous for studies where sample input cannot be upscaled to counterbalance losses. While applying an orthogonal SEC-based enrichment helped to boost crosslink IDs to an overall combined number of >5000 unique XLs on 1692 proteins in the whole cell samples, its tricky to compare yielded coverage to other studies as conditions such as cell type, linker type, sample preparation workflow, LC-MS settings, data analysis software and search settings differ a lot. However, to give an example from another comprehensive study in K562 cells, Yugandhar et al.[35] found 9300 crosslinked residues of which 1268 were inter-protein crosslinks, using DSSO. While this outperforms the numbers seen in this study, they performed extensive SCX fractionation and generated 122 raw files to obtain such an impressive result. In contrast, only 12 raw files yielded the combined result in this study, and of those 5000 unique XLs found (see Fig. 4C), more than 3000 unique XLs on average were found per replicate (in 4 raw files, see Fig. 3D for average numbers and individual results from each replicate). Our data further shows that even without fractionation and using only a single-step affinity enrichment (hence suffering from the least potential sample losses), around 2000 unique crosslinked sites from in vivo crosslinked K562 cells are yielded. Our data further shows that also after SEC fractionation, measurement of only one fraction is sufficient to yield close to 90% of all links found in all fractions which is still a substantial boost of crosslink coverage by around 27% to >2500 unique crosslinks, compared to the 1 step enrichment, with the additional benefit of saved MS measurement time.

We generated a comprehensive proteome-wide PPI network from K562 cells as well as a network from their nuclear extracts, containing 56 (14% of total) novel PPIs not listed in the STRING database. Our data can therefore be used to screen for novel interactors of interest or as a complementary dataset confirming orthogonal biochemical approaches. We additionally present that performing the crosslinking reaction within a cellular sub-compartment (i.e., nucleus) helps to improve the finding of crosslinks on proteins not abundant enough otherwise.

Of note, our attempts to improve predicted protein structures of DDX39 by integrating crosslink data using AlphaLink2 yielded altered protein-complex structures with no improvement in the number of satisfied crosslinks. We hypothesize that the high homology of both protein sequences (DDX39A and B) in automatic sequence alignment yielded suboptimal results. With an altered sequence not identical to the original sequence of DDX39B, predicted structures potentially did not fit equally well to the obtained crosslink data. Furthermore, multiple protein(-complex) conformations present in solution might make it impossible to find a single protein-complex structure satisfying all false-positive crosslinks at a time.

The presented easy-to-implement and comprehensive in vivo crosslink workflow will be highly valuable for the community, including non-cross-link-experts, who can apply this simple workflow to any cell type or enriched subcellular fraction with general wet lab skills. The identified shared PPI network displayed in this study can be further used as a general resource for data mining and will serve as a complementary dataset for (novel) PPIs identified with orthogonal approaches in the future.

## Data availability

The mass spectrometry raw proteomics data, Proteome Discoverer search results, and used fasta files have been deposited to the ProteomeXchange Consortium via the PRIDE[36] partner repository with the dataset identifier PXD061173.MS Amanda 3.0 (https://github.com/hgb-bin-proteomics/MSAmanda) and MS Annika 3.0, related workflow templates used for data analysis (https://github.com/hgb-bin-proteomics/MSAnnika) as well as code to generate XiView compatible exports from Annika data (https://github.com/hgb-bin-proteomics/MSAnnika_exporters/blob/develop/xiViewExporter_msannika.py) are available on GitHub. Source data of all figures is provided as Supplementary Data 1. Predicted structures as given in Fig. 5 are made available as Supplementary Data 2–4.

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

## Acknowledgements

The authors would like to thank Ulrich Hohmann, Julius Brennecke, and Nuno Maulide for their support and fruitful discussions throughout this project. Furthermore, we would like to express our gratitude to Micha Birklbauer for his continuous support using Annika and his help in resolving export and data processing problems. This work was supported by the infrastructure funding 4th call 2022/01 (AT-SCP) of the Austrian Research Promotion Agency (FFG) and the project LS20-079 of the Vienna Science and Technology Fund (WWTF). This work was further funded by the P35045-B project (Grant-DOI 10.55776/P35045) and the F 8801-B Meiosis project (Grant-DOI 10.55776/F88) of the Austrian Science Fund (FWF). F.M. is supported by ESP 566 (Grant-DOI 10.55776/ESP566) of the FWF. This research was funded in whole or in part by the FWF. Research at the IMP is supported by Boehringer Ingelheim. For the purpose of open access, the authors have applied a CC BY public copyright license to any Author Accepted Manuscript version arising from this submission.

## Author contributions

P.B. conducted all crosslink experiments, LC-MS measurements, and wrote the manuscript. L.T. provided K562 cells and generated fluorescence images. F.M. contributed to the 3D model interpretation. M.M. performed data analysis, post processing, interpretation, and wrote the manuscript. K.M.

and M.M. conceptualized and supervised the study. All authors revised and agreed on the paper.

## Competing interests

The authors declare no competing interests.
