## [Transparent Peer Review file · Communications Chemistry]

In vivo crosslinking and effective 2D enrichment for proteome wide interactome studies.

Corresponding Author: Dr Manuel Matzinger

Version 0:

Reviewer comments:

Reviewer #1

(Remarks to the Author)

In this study Brauer et al describe an optimised crosslinking strategy for in vivo studies of protein interactions using the established crosslinked DSBSO. Several studies have been performed previously using DSBSO and have developed methods using this reagent. Here the authors show that combining enrichment via affinity (here using Click reactions) and SEC result in higher number of identified crosslinks. Whilst this is important, it is certainly not unexpected. Indeed others have shown the fractionation subsequent to enrichment results in improved crosslink detection. Our main concern is that the manuscript lacks sufficient detail in certain places, and the biological findings that the authors discuss are not well supported. This study could be reconsidered after major revisions.

- Title. This states the study relates to the nucleosome. It does not – this is a method development study alone. There is no further mention of nucleosome in this manuscript. The title must be amended.
- Move existing Figure 7 so it is Figure 1 and provide a comprehensive figure legend.
- Line 79. Add a comment that this 'background' is not going to mimic an in-cell crosslinking experiment. The relative abundances of crosslinked peptides under these conditions is going to be significantly higher than in any in-cell crosslinking experiment.
- Line 86. What is meant by 'quite sticky' and 'non-adherent behaviour eliminating smearing effects'? An important aspect of this manuscript is the workflow validation and comparison between different methods. However, the details provided are sparse. In this same section (lines 78-96) the authors should also mention briefly the LC-MS methods used.
- Figure 1A-C. The authors must perform a control experiment where they perform enrichment without supplementing their in vitro crosslinked Cas9 sample with HeLa digest (i.e. left panels of A-C) to demonstrate that a similar increase in the number of identified crosslinks is obtained as when they perform enrichment after supplementing with HeLa digest (i.e. right panels of A-C).
- Line 116. More details should be included here (briefly) of the SEC conditions used. The data suggest that 'Fraction 9' is the optimal fraction for analysis. It is unclear to us how any other laboratory will be able to identify their version of 'fraction 9' under their experimental conditions. More detail is needed to provide the reader with an understanding of how they could identify a fraction with high XL abundance without performing significant MS experiments. The authors must show representative SEC chromatograms identifying the position of 'fraction 9' for clarity, and describe the solution and column conditions used in detail.
- Line 139. Why were different database sizes used? What was the hypothesis behind this change in approach. The authors state the smaller database size results in fewer chances for decoy matches – what are the quality of the low scoring matches that are 'new' upon database reduction? Some representative mass spectra would be a good addition to demonstrate that this altered methodology does not result in the additional identifications being of very poor quality.
- Line 166 – 'we made applied a clone'. Check grammar. Was this expressed on a plasmid in the cells or was this a CRISPR modification to the genome? There is no detail of this included in the results or methods sections. More clarification is needed on how this experiment was conducted.
- Line 176. We don't think the authors need to include analysis of the AF model of the tagged protein here. It complicates matters and the authors show it has no effect. Assuming that they remove from their analysis any DDX39-GFP crosslinks we don't see a problem with this.
- Line 180. Are the authors saying that their models are of low quality? This is unclear. pTM is a global indicator of prediction quality – it is not unexpected that the overall pTM score will be dramatically reduced when modelling a GFP tagged version of a native protein because AF will not be able to place the 'domains' of the chimeric proteins in three dimensions. Remove this discussion as it does not add any meaning to the interpretation of the results.
- Line 182. What is the evidence that DDX39A and B is a monomer (presumably some other in vitro studies exist that should be referenced), or form a homo or heterodimer? The pTM score is very low for the complex, as they state, and the authors comment it could be because there are additional components not present e.g. RNA. We find this explanation very unconvincing. If this protein was over-expressed then there could be non-native contacts formed as a result of high local

concentrations in the nucleus – indeed the authors state that others have reported dimerization upon overexpression. Is this artefactual as a result of overexpression? The authors must comment on this and provide additional information to support their modelling conclusions. Indeed, their data (combined from AF and XL), and those in the literature, suggest that these dimers could be non-native. This calls into question if any biological conclusions can be drawn from this analysis.

- Line 181. What is the level of sequence conservation between DDX39A and DDX39B? How confident are the authors that the crosslinks they are seeing are from a heterodimer? Some evidence to support this claim would be helpful.
- Line 193. The authors should generate AF models of DDX39A/B and CHTOP to determine if their detected crosslinks are consistent with structural models.
- Line 220. The authors state that when crosslinking with nuclear extracts, 80 % of crosslink matches were annotated to the nucleus. However, their data are more nuanced than this. Indeed only 36.5 % of matches were to nuclear proteins and 43.5 % of matches were to 'cytosol or nucleus'. Then the authors state as little as 0.5 % or 0.2 % for whole cell or nuclear extracts, respectively, were between proteins from different cellular compartments. What does this mean in cases when a protein could be cytosolic or nuclear? Are more protein-protein crosslinks involving nuclear proteins only detected upon nuclear enrichment?
- Supplementary Tables have not been provided during the review process so we are unable to comment on the claims supported by certain statements.
- Line 413. We are not sure that the authors can claim the suitability of their methods for studies with limited sample input. Indeed a significant protein input is required for the enrichment method proposed.
- In the discussion (line 419) the authors mention the number of replicates performed. Mention of this reproducibility should be included in the results and shown in an appropriate figure.
- Line 432. We are not convinced the authors have discovered any new biology around DDX39 from their analysis. This should be removed or clarified.
- Line 432. The authors state they tried to use alphascreen in the discussion, but provide no support/results for the statements they make.
- Supplementary Figure 3. What are the brown bars? Indeed, throughout many figures have insufficient legends to provide the reader with the necessary information needed to interpret the data presented.

Minor comments.

The manuscript needs thorough proof reading. A few examples where we have found changes are needed are:

- Line 20 – the word pulling is not the correct term. Perhaps replace with 'enrichment'.
- Line 27 – change form to forms.
- Line 44 – define the n2 problem for a more general audience.
- Line 51 – delete 'surprisingly well'
- Line 112 – add 'this' before unexpectedly.
- Line 223. 'and were calculated as co-localization'. What does this mean? Please amend.
- Line 234. change primary to primarily.
- Line 410. Amend guideline to guidelines.
- Line 415. Full stop before 'its tricky' should be a comma.

Reviewer #2

(Remarks to the Author)

Matzinger and colleagues report on the development of an in vivo crosslinking method using the Azide-A-DSBSO linker and a two-step enrichment method. The authors first compared different bead materials for click chemistry enrichment through coupling of the azide group on the linker using Cas9 as a model protein spiked into a HeLa digest as background. The optimized enrichment protocol is then applied to in vivo crosslinking in K562 cells, and the single-step enrichment method is compared to a two-step enrichment where a size-based fractionation step is added as step 2. It is noteworthy that with their SEC setup a large fraction of crosslink products is contained in one fraction, enabling high sample throughput. As an example for a downstream application, the RNA helicase DDX39B is further explored; crosslinks are mapped to predicted structures of monomer and dimer, and the interaction network obtained from crosslinking of whole cells and of nuclear extracts is compared.

Overall, the authors present a streamlined method for efficient enrichment of crosslinking products from the clickable Azide-A-DSBSO linker that should be useful for others in the field. The manuscript is well written and organized, although I have some suggestions that should be considered in a revised version:

Throughout the text: The brand name is Cytiva, not Cytivia, please correct.

Line 68: "we applied DSBSO to the frozen cells". This contradicts the method description (line 280): "frozen cells ... were thawed quickly and resuspended in a crosslinking buffer". Please clarify.

Figure 1B: Why are CSMs and numbers of peptides (not PSMs?) shown in this panel? One data type is redundant, the other not.

Figure 6B: The PPI network is hardly readable even after magnification (labels overlap, poor resolution), maybe there is a better way to present this?

Line 268: Is there a reference for the REAP protocol?

Line 284-290: It can be argued that quenching may not be needed in this setup, but from the protocol further below the reagent is not removed "immediately" if sonication and centrifugation follow, which likely takes at least 30 min. Also, if the benzonase really makes a difference for crosslinking cells, then the cells must have been at least partially lysed by then because the nuclease would not be internalized?

Line 291-294: The assumption that 1 mg protein yields 1 mg peptides is probably overly optimistic?

Line 433: AlphaLink2 appears out of nowhere here. At least a reference should be added.

Wording:

Figure 3, legend: "XL unique XL sites" (twice).

Line 166: "we made applied a clone"

Figure 4F, heading on the right panel: "Measured possible XL of overlong on native dimer"

Line 237: "able to go cover"

Reviewer #3

(Remarks to the Author)

I co-reviewed this manuscript with one of the reviewers who provided the listed reports. This is part of a Communications Chemistry initiative to facilitate training in peer review and to provide appropriate recognition for Early Career Researchers who co-review manuscripts.

Reviewer #4

(Remarks to the Author)

Inter-peptide crosslinks, which provide most valuable distance information, constitute minor fraction of the total peptides produced by enzymatic digestion of the crosslinked protein samples. Several approaches were proposed to enrich such crosslinked peptides, including affinity enrichment, ion-exchange and size exclusion chromatographies. In the submitted manuscript authors propose to combine affinity enrichment of the crosslinks with size exclusion chromatography (SEC). They illustrated improvement in the number of the detected inter-peptide crosslinks, even using single SEC fraction. The strategy is reasonable, and the proposed approach deserves publishing in Chem. Comm. after minor revision.

Major critiques:

1. The title does not really reflect content of the manuscript. 2D enrichment of what? Apparently, the method is proposed for general applications, not only for study nucleosomes.
2. Reported numbers of crosslinks need to be clarified in figure legends and better discussed in text.
3. Final amounts of starting material used needs to be described in the methods and included in the discussion of the obtained crosslink numbers.
4. Terminology needs to be cleared up. The reviewer is not a fan of "XL" (extra-large) designation of the crosslinks, but eventually it is authors choice. In figures and figure legends, it refers to inter-peptide crosslinks? CSM refers to inter-peptide crosslinks? "Inter XL" refers to inter-protein?

Minor critiques:

Lane 83.

CSM, crosslink spectral matches.

Lane 97.

Needs to be mentioned, crosslinks numbers refer only to Cas9 crosslinks.

Lane 98.

Both figures 1 and 2 are proof-of-principle.

Lane 132.

Linear peptides needs to be clarified detected after affinity enrichment. See major critique 2. Why no inter-protein crosslinks at all were detected without SEC? Why twice more monolinks detected with SEC step? Why whole cell and nuclei produced the same numbers of inter-peptide crosslinks? Why linear peptide numbers are the same with and without SEC?

Lane 158.

Figure title should be descriptive.

Version 1:

Reviewer comments:

Reviewer #1

(Remarks to the Author)

We appreciate the thorough nature of the changes made by the authors to their manuscript.

The title now better reflects the major claims of the updated manuscript: which includes a protocol to map the interactome of subcellular environments to elucidate biological understanding of nuclear proteins. The novelty of characterising and optimising the workflow to produce this data is of real importance to the field and will hopefully encourage more in-cell crosslinking experiments in the future. The authors have included more in-depth methods and more thorough explanations of experimental designs in this revision, making the approach presented accessible to other labs. We now recommend publication of the revised manuscript.

Reviewer #2

(Remarks to the Author)

With this revised version, the authors have addressed all my comments appropriately. I have no further requests.

Reviewer #3

(Remarks to the Author)

The major claims of the updated manuscript are of the description of a protocol to map the interactome of subcellular environments to elucidate biological understanding of nuclear proteins. The novelty of the biological understanding is not fully conclusive as stated by the authors, however, the work required to fully understand the mechanisms of DDX39 would not be in the scope of this study. The conclusions drawn about DDX39's complexity may inspire further work into this complex and its structural behavior in cells. The novelty of characterising and optimising the workflow to produce this data is of real importance to the field and will hopefully encourage more in-cell crosslinking experiments in the future. After the addition of more in-depth methods and explanation of experimental design makes the data presented reproducible.

Reviewer #4

(Remarks to the Author)

OK now.

-N3 is not charged group.

Reviewer #1 (Remarks to the Author):

In this study Brauer et al describe an optimised crosslinking strategy for in vivo studies of protein interactions using the established crosslinked DSBSO. Several studies have been performed previously using DSBSO and have developed methods using this reagent. Here the authors show that combining enrichment via affinity (here using Click reactions) and SEC result in higher number of identified crosslinks. Whilst this is important, it is certainly not unexpected. Indeed others have shown the fractionation subsequent to enrichment results in improved crosslink detection. Our main concern is that the manuscript lacks sufficient detail in certain places, and the biological findings that the authors discuss are not well supported. This study could be reconsidered after major revisions.

We thank the reviewer 1 & their co-reviewer 3 for carefully evaluating our work and giving valuable input on improvement for our study. We considered all raised questions, provide detailed answers below and added additional experimental data as suggested in our revised manuscript.

- Title. This states the study relates to the nucleosome. It does not – this is a method development study alone. There is no further mention of nucleosome in this manuscript. The title must be amended.

We apologize that our title was misleading. We related nucleosome to the fact that we yield a PPI network from proteins within the nucleus as we used both whole cells and enriched nuclei for crosslinking. We agree that the focus of this study is on method development and hence rephrased the title to a more general “In vivo crosslinking and effective 2D enrichment for proteome wide interactome studies.” to express what the workflow is meant to be used for.

- Move existing Figure 7 so it is Figure 1 and provide a comprehensive figure legend.

We followed the reviewer’s suggestion and moved Figure 7, containing the workflow, from the methods section to the beginning of the results section, now being Figure 1. We further extended the figure legend with full details of the workflow still being displayed in the Methods section of the manuscript.

- Line 79. Add a comment that this ‘background’ is not going to mimic an in-cell crosslinking experiment. The relative abundances of crosslinked peptides under these conditions is going to be significantly higher than in any in-cell crosslinking experiment.

We agree with the reviewer and add an additional statement to avoid potential confusion: “...Of note, the relative abundance of non-crosslinked material in a real *in cellulo* experiment is even higher, hence this is only the first step using a simple model system to evaluate enrichment performance. “

Furthermore, in our revised manuscript we added a new dataset including enrichment controls without spiking and spiking at a higher Cas9:HeLa ratio of 1:80, which we believe is significantly closer to a real-world scenario. This experiment shows that we can identify only ~30-50% (depending on bead type) of the maximal crosslink-ID numbers after enrichment and without spiking, while it also proves that recovery of crosslinked peptides is possible from that high background level and an otherwise (without enrichment) useless sample were no crosslinks could be identified anymore (see Figure 2 A-C in the revised manuscript).

- Line 86. What is meant by ‘quite sticky’ and ‘non-adherent behaviour eliminating smearing effects’? An important aspect of this manuscript is the workflow validation and comparison between different methods. However, the details provided are sparse. In this same section (lines 78-96) the authors should also mention briefly the LC-MS methods used.

We apologize for the confusion caused and rephrased that section to improve clarity. We further followed the suggestion of the reviewer and added brief details of LC-MS methods used in the respective figure legend as we think this improves convenience in finding the key settings when inspecting the results. We appreciate the reviewers comment on sparse details (i.e. on LC-MS settings) provided within the results section. The authors would like to refer the reviewer to the Methods section, where full details are provided. Providing full method details in their respective section rather than in the results section is commonly done in proteomics method development papers to the best of our knowledge. Furthermore, all raw files, which include the used method, and which can be directly loaded to generate a new method on Thermo instruments, are provided in the PRIDE repository.

- Figure 1A-C. The authors must perform a control experiment where they perform enrichment without supplementing their in vitro crosslinked Cas9 sample with HeLa digest (i.e. left panels of A-C) to demonstrate that a similar increase in the number of identified crosslinks is obtained as when they perform enrichment after supplementing with HeLa digest (i.e. right panels of A-C).

We thank the reviewer for their valuable suggestion, clearly improving the quality of our study. In line with the reviewer’s suggestion, we performed such a control experiment, which is now shown in Figure 2A-C of the revised manuscript. Although the relative increase in crosslinks rises with increasing background levels as expected (as crosslink numbers tend towards zero with increasing background level), we still observed an increase in crosslink ID numbers of 169 -314% (depending on bead type) when working with Cas9 as single purified protein.

Of note, yielded crosslink numbers are generally higher than shown in the original experiment which might be attributed to a different batch of magnetic beads and DSBSO used, with our old batch found completely hydrolyzed and not useable for the revision.

- Line 116. More details should be included here (briefly) of the SEC conditions used. The data suggest that ‘Fraction 9’ is the optimal fraction for analysis. It is unclear to us how any other laboratory will be able to identify their version of ‘fraction 9’ under their experimental conditions. More detail is needed to provide the reader with an understanding of how they could identify a fraction with high XL abundance without performing significant MS experiments. The authors must show representative SEC chromatograms identifying the position of ‘fraction 9’ for clarity, and describe the solution and column conditions used in detail.

Regarding the SEC conditions we would like to refer the reviewer to the Methods section of the manuscript which already contains used column, gradient and buffer composition.

We appreciate and agree on the reviewers comment on the difficulty of reproducing the exact same conditions in another lab and added a representative SEC chromatogram as Supplemental Figure 3 as suggested. We further extended the respective results section as follows:

“However, a sufficient separation of monolinked from crosslinked peptides was facilitated via SEC specifically in the early fractions, containing more and larger crosslinked peptides. In these early fractions the CSM/monolink ratio was improved to >6 and drops to <0.2 for later eluting fractions (Supplemental Figure 2). Of note, a complete separation of monolinked from crosslinked peptides was

not expected and our results successfully reproduce previous reports for SEC based crosslink enrichment.¹² In this context, we noticed that the vast majority of crosslink IDs, namely ~87% & ~90% when crosslinking the entire cell or the enriched nuclei, respectively, were found in only one SEC fraction (**Error! Reference source not found.A-B**). To select for the best performing fraction, measurement of all fractions of potential interest is needed at least for one replicate though. For repetitions and experiments with similar settings experience from that initial measurement can be used to make a fast decision on fractions of interest for measurement. For initial selection of fractions to be subjected to LC-MS we used the obtained UV trace from SEC and selected early fractions starting at the point where the UV signal started to rise and ending where a huge background peak appears (Supplemental Figure 3).“

With this we hope to better explain how we selected fractions of interest aiming for others being able to reproduce our steps with their specific settings. This selection process is backed up by the data provided in Figure 4 A & B as well as Supplemental Figure 2.

- Line 139. Why were different database sizes used? What was the hypothesis behind this change in approach. The authors state the smaller database size results in fewer chances for decoy matches – what are the quality of the low scoring matches that are ‘new’ upon database reduction? Some representative mass spectra would be a good addition to demonstrate that this altered methodology does not result in the additional identifications being of very poor quality.

We thank the reviewer for having raised this question as we are convinced, the additional details we have added in response significantly improve the quality of our manuscript!

The authors used different database sizes to optimize the database search finding a good balance between huge (proteome wide) searches with a potentially too large decoy database and huge search space and too small databases potentially not allowing for proper target-decoy statistics anymore. The assumption that too large databases might cause problems (next to extended search times) is further based on the number of redundant sequences within the database, which is increased with increasing size. This leads to ambiguous crosslink IDs as described by the Bruce lab some years ago (Chavez, J. D et al. Protein interactions, post-translational modifications and topologies in human cells. *Mol. Cell. Proteom.* **12**, 1451–1467 (2013))

We found that using a database containing all proteins previously found in a shotgun analysis (hence all proteins abundant enough to likely form any crosslink) yielded most crosslink IDs. It further improved coverage for our protein of interest, DDX39B. We checked in an earlier study if there is an influence on the quality of the identification and found that this is not the case for MS Annika, which was used as search engine in this study. (Matzinger et al., Mimicked synthetic ribosomal protein complex for benchmarking crosslinking mass spectrometry workflows. *Nat. Commun.* **13**, 3975 (2022) <https://www.nature.com/articles/s41467-022-31701-w/figures/6>).

In response to this question we further checked for differences in the crosslink-score distribution obtained from MS Annika running with a 1% FDR filter (new Supplemental Figure 4 A, B) and found that there is indeed a slight shift towards lower scores accepted for 1% FDR with databases smaller than the full human proteome. This likely explained improved XL numbers. The relative fraction of decoy vs target-target hits remains at a comparable level though (Supplemental Figure 4 C) as does the spectrum quality: In line with the reviewer’s suggestion, we added representative mass spectra of the worst scored XL-ID for each database size in our new Supplemental Figure 5, showing comparable quality for searches towards the largest and smallest database size used.

We extended our results section with a short discussion and more details.

- Line 166 – ‘we made applied a clone’. Check grammar. Was this expressed on a plasmid in the cells or was this a CRISPR modification to the genome? There is no detail of this included in the results or methods sections. More clarification is needed on how this experiment was conducted.

We thank the reviewer for pointing us to this missing detail. We extended the methods section to explain how the clone was generated and corrected the grammar error.

- Line 176. We don’t think the authors need to include analysis of the AF model of the tagged protein here. It complicates matters and the authors show it has no effect. Assuming that they remove from their analysis any DDX39-GFP crosslinks we don’t see a problem with this.

We agree with the reviewer, that there seems no effect on the DDX39B structure based on presence or absence of the tag included to the model. We would like to argue, that this information is already of interest: Given the comparable size of DDX39B (438 amino acids) vs the FKBP-GFP tag (386 amino acids) a potential influence of the tag to the structure is a major concern relevant also to future studies as it might also influence biological function and hence validity of experiments performed with that specific K562 clone (or with tagged DDX39B in general). We therefore decided to keep both models within figure 5 of the revised manuscript.

- Line 180. Are the authors saying that their models are of low quality? This is unclear. pTM is a global indicator of prediction quality – it is not unexpected that the overall pTM score will be dramatically reduced when modelling a GFP tagged version of a native protein because AF will not be able to place the ‘domains’ of the chimeric proteins in three dimensions. Remove this discussion as it does not add any meaning to the interpretation of the results.

We thank the reviewer for their advice and removed the discussion of the pTM score as suggested.

- Line 182. What is the evidence that DDX39A and B is a monomer (presumably some other in vitro studies exist that should be referenced), or form a homo or heterodimer? The pTM score is very low for the complex, as they state, and the authors comment it could be because there are additional components not present e.g. RNA. We find this explanation very unconvincing. If this protein was over-expressed then there could be non-native contacts formed as a result of high local concentrations in the nucleus – indeed the authors state that others have reported dimerization upon overexpression. Is this artefactual as a result of overexpression? The authors must comment on this and provide additional information to support their modelling conclusions. Indeed, their data (combined from AF and XL), and those in the literature, suggest that these dimers could be non-native. This calls into question if any biological conclusions can be drawn from this analysis.

We thank the reviewer for carefully evaluating possible explanations for mono- or dimeric- forms being present under native conditions. We agree that within the study, focusing on method development, drawing solid biological conclusions is not possible. Nevertheless, we think that our study showcases how hints for the presence of dimeric forms can be found by combining XL and AF prediction data. We hence adopted our results section to make clear that the statements made on biology of DDX39 are possible explanations that would require additional and orthogonal experiments in a follow up study to draw solid conclusions.

The used cell line is a CRISPR tagged one excluding the possibility of overexpression. We included additional data from western blotting that suggests, that tagging makes the protein less stable, resulting in slightly reduced protein levels compared to the wild-type, excluding artificial crosslinks due to protein levels higher than native (Supplemental Figure 7 in revised manuscript).

Next to the cited literature on overexpressed DDX39A, referred to here by the reviewer, no study on endogenous protein levels of DDX39A and B is known to the authors. A study from the Plaschka lab (<https://elifesciences.org/articles/61503>), shows the essential transcription–export complex (TREX) that includes 4 UAP56/DDX39 molecules, but did not investigate DDX39 alone, or differences between A/B forms as both are supposed to have the same function. In their structure, however, the UAP56/DDX39 molecules are positioned very close to each other, with the distance between the C-terminal RecA lobe of one UAP56 and the N-terminal RecA lobe of the neighboring one in the 20Å range, which can explain crosslinks within this complex as well. The exact link position found within this study is not covered by the structure (PDB: 7APK) of that publication though. There is further literature, showing that once UAP56 is bound to RNA it dissociates from THO (<https://www.biorxiv.org/content/10.1101/2024.03.24.586400v2>), resulting in some flexibility in how neighboring UAP56 molecules are positioned once released from THO. **To sum up:** To the best of our knowledge, there is no published hard evidence for UAP56 dimerization, but there is evidence for UAP56 molecules being in very close proximity, which is in line with the hints we received from our crosslinking data.

- Line 181. What is the level of sequence conservation between DDX39A and DDX39B? How confident are the authors that the crosslinks they are seeing are from a heterodimer? Some evidence to support this claim would be helpful.

The sequence identity of DDX39A and B is 89.95% when using BLAST® (<https://blast.ncbi.nlm.nih.gov>) to compare both sequences. The used search engine MS Annika within Proteome Discoverer reports >1 protein annotations for all peptides/CSMs or XLs in case the peptide sequence behind is ambiguous, while only a single protein annotation is reported for unambiguous hits. With regard to the question, we found peptides of 6 ambiguous sequences connected to 30 unique crosslinked residue pairs and 383 CSM (where peptide A and B of the crosslink is ambiguous). For DDX39A peptides from 8 unambiguous sequences connected to 27 unique residue crosslinked residue pairs and 223 CSM were found (where at least one of both connected peptides is unambiguous). For DDX39B peptides from 6 unambiguous sequences were found connected to 11 unique crosslinked residue pairs and 206 CSM (where at least one of both connected peptides is unambiguous). We further found an unambiguous crosslink from DDX39 A and B to CHTOP respectively. We however did not find exclusively ambiguous DDX39A/B crosslink, which is why its existence is only further supported from the results obtained from mapping the crosslinks to the predicted structures. All found links can be found within Supplemental Table 3.

Long story short: Our data provides evidence for the existence of both proteins detected and crosslinked within the sample.

We added an additional sequence coverage overview as new supplemental figure 6 and extended the respective results section with the above explained details.

- Line 193. The authors should generate AF models of DDX39A/B and CHTOP to determine if their detected crosslinks are consistent with structural models.

We followed the reviewer's suggestion and generated respective AF structures, now part of the extended Figure 6. The found inter-protein connections of DDX39A or B to CHTOP well align with the expected distance threshold. We also plotted unambiguous intraprotein links found on DDX39A/B. No intraprotein link was found on CHTOP. Only a single unambiguous intraprotein link was found on DDX39B, that also aligns with the expected distance. 4/10 unambiguous intraprotein crosslinks on

DDX39A are overlength and using the XL data to guide structure prediction resulted in only minor changes of the crosslink lengths (see question for line 432).

We extended Figure 6 with the respective predicted structures and discussed the results in the respective results section of the revised manuscript.

- Line 220. The authors state that when crosslinking with nuclear extracts, 80 % of crosslink matches were annotated to the nucleus. However, their data are more nuanced than this. Indeed only 36.5 % of matches were to nuclear proteins and 43.5 % of matches were to 'cytosol or nucleus'. Then the authors state as little as 0.5 % or 0.2 % for whole cell or nuclear extracts, respectively, were between proteins from different cellular compartments. What does this mean in cases when a protein could be cytosolic or nuclear? Are more protein-protein crosslinks involving nuclear proteins only detected upon nuclear enrichment?

Yes, the reviewer is correct: The given 80% nuclear proteins refer to a scenario where all proteins with ambiguous GO-annotation are counted as nuclear. Since this ambiguous annotation means they were reported in literature to be present both, in the nucleus and in the cytosol, we think it is valid to assume that such crosslinks were formed from and to proteins within the same cellular compartment (in that case nucleus) as the alternative is much less likely (due to enrichment for nuclear fraction as well as the crosslinker being present only in one compartment at once while reacting with proteins).

To make clearer to the readers how we decided on dealing with ambiguous GO annotations we extended the explanation statement in the results section as follows:

"Proteins with ambiguous localization annotation, that are present within the nucleus and the cytosol according to their GO-annotations are depicted as such in Figure 7A and were counted as co-localization."

To the second part of the question: Yes, more proteins involving nuclear proteins are detected upon nuclear enrichment when looking only to proteins with unambiguous localization annotation. Within the whole cell dataset ~18 % of all crosslinks was involving nuclear proteins, which rises to ~37% in the dataset from nuclear extracts. In line, the opposite is true when looking to proteins localized unambiguously in the cytosol; here the fraction is lowered from 30% to 14% in whole cell vs nuclear extract data respectively (data shown in Figure 7A).

- Supplementary Tables have not been provided during the review process so we are unable to comment on the claims supported by certain statements.

We apologize for the tables not being provided and ensured they are uploaded with the revised version of the manuscript.

- Line 413. We are not sure that the authors can claim the suitability of their methods for studies with limited sample input. Indeed a significant protein input is required for the enrichment method proposed.

Agreed and rephrased to

"This is especially advantageous for studies where sample input cannot be upscaled to counterbalance losses." to avoid confusion.

- In the discussion (line 419) the authors mention the number of replicates performed. Mention of this reproducibility should be included in the results and shown in an appropriate figure.

We added a cross reference to the correlating figures within the results section in the respective paragraph of the discussion to improve clarity. We ensured that, wherever averages from replicate measurements are shown throughout the manuscript, standard deviations as well as individual values from each replicate are shown.

- Line 432. We are not convinced the authors have discovered any new biology around DDX39 from their analysis. This should be removed or clarified.

We checked the discussion and specifically the referenced paragraph. It contains a discussion about problems with alfaLink2 (see next question) and potential problems with modelling in general. We did not find any statement about new biology around DDX39. We checked the same in the results section, where we already included a statement saying that DDX39A&B dimerization as well as its interaction to CHTOP was previously reported.

- Line 432. The authors state they tried to use alfaLink2 in the discussion, but provide no support/results for the statements they make.

We agree with the reviewer's opinion. In short, using AlphaLink2 in our hands and for this specific protein-complex did not improve the fraction of satisfied crosslinks. In response to the reviewer's question we adopted the discussion to improve clarity and added Figure 6 C & D where we showcase the application of AlphaLink2 on the DDX39A or B - CHTOP complex. We further expanded our results section discussion the result.

- Supplementary Figure 3. What are the brown bars? Indeed, throughout many figures have insufficient legends to provide the reader with the necessary information needed to interpret the data presented.

We apologize this graph has caused confusion. Within that figure (now Supplemental Figure 7) overlapping regions of green and red bars appear brown due to the transparency effect. Transparency is needed to see the actual distribution of both nuclear and whole cell obtained crosslinks properly.

We added a statement to the figure legend and checked all figure legends in the manuscript for potential missing information. We thereby aimed to balance well between providing sufficient information without being repetitive or providing exhaustive information in the legend (i.e. full method & processing details of graph generation) as we think this would hamper effective reading and understanding of the manuscript.

Minor comments.

The manuscript needs thorough proof reading. A few examples where we have found changes are needed are:

- Line 20 – the word pulling is not the correct term. Perhaps replace with 'enrichment'.

implemented

- Line 27 – change form to forms.

implemented

- Line 44 – define the n2 problem for a more general audience.

implemented

- Line 51 – delete ‘surprisingly well’

Added a statement why this is surprising to us (as we are trained chemists)

- Line 112 – add ‘this’ before unexpectedly.

implemented

- Line 223. ‘and were calculated as co-localization’. What does this mean? Please amend.

The text there says: “counted as co-localization” (not calculated), which seems clear to us in the context of the full sentence.

- Line 234. change primary to primarily.

implemented

- Line 410. Amend guideline to guidelines.

implemented

- Line 415. Full stop before ‘its tricky’ should be a comma.

implemented

Reviewer #2 (Remarks to the Author):

Matzinger and colleagues report on the development of an in vivo crosslinking method using the Azide-A-DSBSO linker and a two-step enrichment method. The authors first compared different bead materials for click chemistry enrichment through coupling of the azide group on the linker using Cas9 as a model protein spiked into a HeLa digest as background. The optimized enrichment protocol is then applied to in vivo crosslinking in K562 cells, and the single-step enrichment method is compared to a two-step enrichment where a size-based fractionation step is added as step 2. It is noteworthy that with their SEC setup a large fraction of crosslink products is contained in one fraction, enabling high sample throughput. As an example for a downstream application, the RNA helicase DDX39B is further explored; crosslinks are mapped to predicted structures of monomer and dimer, and the interaction network obtained from crosslinking of whole cells and of nuclear extracts is compared.

Overall, the authors present a streamlined method for efficient enrichment of crosslinking products from the clickable Azide-A-DSBSO linker that should be useful for others in the field. The manuscript is well written and organized, although I have some suggestions that should be considered in a revised version:

We thank the reviewer for their detailed evaluation and positive feedback. Please find our reply to the raised questions below.

Throughout the text: The brand name is Cytiva, not Cytivia, please correct.

We thank the reviewer for making us aware of this typo and corrected it throughout the manuscript.

Line 68: "we applied DSBSO to the frozen cells". This contradicts the method description (line 280): "frozen cells ... were thawed quickly and resuspended in a crosslinking buffer". Please clarify.

We thank the reviewer for pointing us to this inconsistency in phrasing. Frozen cells were resuspended in the crosslinking buffer and thereby they thawed. Immediately thereafter the crosslinker was added. We adopted the description to improve clarity and ensured consistency throughout the manuscript.

Figure 1B: Why are CSMs and numbers of peptides (not PSMs?) shown in this panel? One data type is redundant, the other not.

We apologize for the confusion caused here. Monolinked peptides referred to PSM with a DSBSO modification, hence also a redundant data type.

We applied a rephrasing to "monolink PSMs" and added a definition in the figure legend. We updated all figures where "monolink PSMs" are shown respectively.

Figure 6B: The PPI network is hardly readable even after magnification (labels overlap, poor resolution), maybe there is a better way to present this?

We agree with the reviewer that this graph is hard to interpret. It was a direct export from xiview, used for all other crosslink visualizations as well but not optimal for such large networks. We replaced that figure with a visualization of the same PPI network generated from Cytoscape. The

overview shows the general size and degree of connectivity of the network but cannot give all details for all proteins. Therefore, we decided to zoom into selected subnetworks. Within those we annotated correlating gene names and performed a gene enrichment analysis to give an impression which functional networks were found within the dataset. We further adopted Supplemental Figure 8 in the same way, for the network from whole cells.

Line 268: Is there a reference for the REAP protocol?

We added the reference to the revised manuscript. (Suzuki et al., BMC Research Notes, 3, 294, 2010)

Line 284-290: It can be argued that quenching may not be needed in this setup, but from the protocol further below the reagent is not removed "immediately" if sonication and centrifugation follow, which likely takes at least 30 min. Also, if the benzonase really makes a difference for crosslinking cells, then the cells must have been at least partially lysed by then because the nuclease would not be internalized?

We agree with the reviewer that excess crosslinker was not removed immediately but only after centrifugation. We removed the word from the text.

Benzonase is added routinely to cellular sample preps in our lab, but its effect for this specific workflow was not evaluated within this study. We assumed at least some degree of a permeabilized membrane at that stage reasoned by the freeze thaw cycle and potentially the shaking at 100 rpm during incubation.

Line 291-294: The assumption that 1 mg protein yields 1 mg peptides is probably overly optimistic?

We agree with the reviewers that this would assume a lossless FASP. We rephrased to: "The peptides eluted from these filters were again pooled to a single tube per replicate."

Line 433: AlphaLink2 appears out of nowhere here. At least a reference should be added.

We extended our results section with Figure 6 C-D, showing results originating from AlphaLink2, and expanded the results section with a description. We further added its citation and added details for Alpha Link 2 to the methods part of the revised manuscript.

Wording:

Figure 3, legend: "XL unique XL sites" (twice). corrected

Line 166: "we made applied a clone" corrected

Figure 4F, heading on the right panel: "Measured possible XL of overlong on native dimer" corrected

Line 237: "able to go cover" corrected

Reviewer #3 (Remarks to the Author):

I co-reviewed this manuscript with one of the reviewers who provided the listed reports. This is part of a Communications Chemistry initiative to facilitate training in peer review and to provide appropriate recognition for Early Career Researchers who co-review manuscripts.

Reviewer #4 (Remarks to the Author):

Inter-peptide crosslinks, which provide most valuable distance information, constitute minor fraction of the total peptides produced by enzymatic digestion of the crosslinked protein samples. Several approaches were proposed to enrich such crosslinked peptides, including affinity enrichment, ion-exchange and size exclusion chromatographies. In the submitted manuscript authors propose to combine affinity enrichment of the crosslinks with size exclusion chromatography (SEC). They illustrated improvement in the number of the detected inter-peptide crosslinks, even using single SEC fraction. The strategy is reasonable, and the proposed approach deserves publishing in Chem. Comm. after minor revision.

We thank the reviewer for their detailed evaluation and positive feedback. Please find our reply to the raised questions below.

Major critiques:

1. The title does not really reflect content of the manuscript. 2D enrichment of what? Apparently, the method is proposed for general applications, not only for study nucleosomes.

We appreciate the reviewer's comment and agree, that the method is intended for general useability within proteome wide crosslink studies. In line also with the suggestion of reviewer 1, we adopted the title accordingly.

2. Reported numbers of crosslinks need to be clarified in figure legends and better discussed in text.

We apologize for confusion caused here. As a similar question was asked by reviewer 1, we rephrased monolinks to monolink PSMs and detailed the definition of all crosslink numbers in the figure legends of main and supplemental figures.

3. Final amounts of starting material used needs to be described in the methods and included in the discussion of the obtained crosslink numbers.

We thank the reviewer for their attention to this detail and adopted the methods section accordingly.

For spike-in experiments using Cas9 as model protein, we used 20 µg Cas9 as starting material for the workflow. For in cell crosslinking or crosslinking of nuclear extracts we used 2E8 K562 cells originating from 120 mL growing media each as starting material. After crosslinking & lysis a Bradford assay was performed to estimate the total protein content. For the following digestion and enrichment protocol the obtained crosslinked lysate was split into equal parts of 1 mg starting material each, to ensure full compatibility.

4. Terminology needs to be cleared up. The reviewer is not a fan of “XL” (extra-large) designation of the crosslinks, but eventually it is authors choice. In figures and figure legends, it refers to inter-peptide crosslinks? CSM refers to inter-peptide crosslinks? “Inter XL” refers to inter-protein?

We agree with the reviewer that different abbreviations for crosslink are used in the field. In line with the abbreviation used in all previous crosslink-related publications of the corresponding author, also within the nature publishing group, we decided to stick to XL, for cross link and appreciate the reviewers understanding.

Within this study terminology as used by the used search engine MS Annika is applied. XL refers to a unique crosslinked residue pair within proteins (i.e. position xx of Protein A to position yy of Protein A or B), with inter always referring to inter-protein and intra meaning it's a crosslink connection within the same protein. CSM refers to redundant sequence matches of crosslinked peptides. We always report total CSM numbers within this study, no differentiating between intra- or inter-peptide CSMs.

We added this definition upon first usage of the abbreviations in lines 94-99 of the revised manuscript to improve clarity to the readers.

Minor critiques:

Lane 83.

CSM, crosslink spectral matches.

Adopted

Lane 97.

Needs to be mentioned, crosslinks numbers refer only to Cas9 crosslinks.

Adopted, added directly within figure legend

Lane 98.

Both figures 1 and 2 are proof-of-principle.

Yes, the first one includes benchmarking of bead types and titration of bead amount needed, the second focusses on the impact of C18 cleanup and reachable crossing coverage in whole cells vs nuclear extracts.

Lane 132.

Linear peptides needs to be clarified detected after affinity enrichment. See major critique 2. Why no inter-protein crosslinks at all were detected without SEC? Why twice more monolinks detected with SEC step? Why whole cell and nuclei produced the same numbers of inter-peptide crosslinks? Why linear peptide numbers are the same with and without SEC?

We thank the reviewer for pointing to this potentially confusing annotation. As already outlined in the reply for critique 2, we clarified all annotations. As for this figure, now Figure 3A vs D, intra-

protein crosslinks have been found with and without SEC as expected. The original figure showed intra- and inter-protein XLs separately only for the SEC panel though, which is also why no inter-, intra- annotation was given in the figure. To improve clarity and make all panels of Figure 3 uniform, we now display inter- and intra-protein XLs for the data with and without SEC.

Monolinks as well as linear peptides are unequally distributed over the SEC fractions (see Supplemental Figure 2, discussion within the results section “Proof of principle study in K562 cells”). Due to the reduced complexity and increased total measurement time, more monolinks were detected & more crosslinks when summarizing data from all measured SEC fractions.

The number of CSMs identified in whole cells (3998) is indeed very close to the number of CSMs identified in nuclear extracts (3889). As these numbers even result from 3 replicates each, with very low standard deviation, we attribute this to high general reproducibility of the used workflow.

Lane 158.

Figure title should be descriptive.

Adopted

We thank the reviewers for their final evaluation of our manuscript and their positive feedback. The reviewers' highly appreciated input helped to clearly improve the quality of the revised manuscript. Please find our response to the remaining comment of Reviewer 4 below.

Reviewer #1 (Remarks to the Author):

We appreciate the thorough nature of the changes made by the authors to their manuscript. The title now better reflects the major claims of the updated manuscript: which includes a protocol to map the interactome of subcellular environments to elucidate biological understanding of nuclear proteins. The novelty of characterising and optimising the workflow to produce this data is of real importance to the field and will hopefully encourage more in-cell crosslinking experiments in the future. The authors have included more in-depth methods and more thorough explanations of experimental designs in this revision, making the approach presented accessible to other labs. We now recommend publication of the revised manuscript.

Reviewer #2 (Remarks to the Author):

With this revised version, the authors have addressed all my comments appropriately. I have no further requests.

Reviewer #3 (Remarks to the Author):

The major claims of the updated manuscript are of the description of a protocol to map the interactome of subcellular environments to elucidate biological understanding of nuclear proteins. The novelty of the biological understanding is not fully conclusive as stated by the authors, however, the work required to fully understand the mechanisms of DDX39 would not be in the scope of this study. The conclusions drawn about DDX39's complexity may inspire further work into this complex and its structural behavior in cells. The novelty of characterising and optimising the workflow to produce this data is of real importance to the field and will hopefully encourage more in-cell crosslinking experiments in the future. After the addition of more in-depth methods and explanation of experimental design makes the data presented reproducible.

Reviewer #4 (Remarks to the Author):

OK now.

-N3 is not charged group.

We apologize for the confusion caused. The reviewer is of course correct, the net charge of the N3 side chain is 0. However, it carries a negative and a positive charged nitrogen atom (summing up to zero charge), stabilized by polar resonance structures. (see e.g. <https://doi.org/10.3390/molecules27123716>)

We rephrased the introduction as follows to enhance clarity.

“..To address these challenges, we use disuccinimidyl bis-sulfoxide (DSBSO), a MS cleavable, enrichable linker that was shown to be surprisingly well membrane permeable, given the charged nitrogen atoms inside the net neutral azide residue”